# In vivo analysis reveals that ATP-hydrolysis couples remodeling to SWI/SNF release from chromatin

Ben C Tilly[1]*, Gillian E Chalkley[1], Jan A van der Knaap[1], Yuri M Moshkin[1], Tsung Wai Kan[1], Dick HW Dekkers[1,2], Jeroen AA Demmers[1,2], C Peter Verrijzer[1]*

[1]Department of Biochemistry, Rotterdam, Netherlands; [2]Proteomics Center, Erasmus University Medical Center, Rotterdam, Netherlands

**Abstract** ATP-dependent chromatin remodelers control the accessibility of genomic DNA through nucleosome mobilization. However, the dynamics of genome exploration by remodelers, and the role of ATP hydrolysis in this process remain unclear. We used live-cell imaging of *Drosophila* polytene nuclei to monitor Brahma (BRM) remodeler interactions with its chromosomal targets. In parallel, we measured local chromatin condensation and its effect on BRM association. Surprisingly, only a small portion of BRM is bound to chromatin at any given time. BRM binds decondensed chromatin but is excluded from condensed chromatin, limiting its genomic search space. BRM-chromatin interactions are highly dynamic, whereas histone-exchange is limited and much slower. Intriguingly, loss of ATP hydrolysis enhanced chromatin retention and clustering of BRM, which was associated with reduced histone turnover. Thus, ATP hydrolysis couples nucleosome remodeling to remodeler release, driving a continuous transient probing of the genome.

*For correspondence:
b.tilly@erasmusmc.nl (BCT);
c.verrijzer@erasmusmc.nl (CPV)

**Competing interests:** The authors declare that no competing interests exist.

## Introduction

ATP-dependent chromatin remodeling enzymes (remodelers) alter the structure and organization of nucleosomes, the basic building blocks of eukaryotic chromatin (*Becker and Workman, 2013*; *Bracken et al., 2019*; *Clapier et al., 2017*; *Jungblut et al., 2020*; *Sundaramoorthy and Owen-Hughes, 2020*). Core nucleosomes comprise 147 bp of DNA wrapped around a histone protein octamer formed by a H3-H4 tetramer that is flanked on either side by a H2A-H2B dimer. Remodelers modulate the accessibility of regulatory DNA elements by unwrapping or sliding nucleosomes, thereby presenting a fundamental level of control of transcription and other processes that require access to DNA (*Brahma and Henikoff, 2020*). Remodelers are large, multi-subunit complexes that harbor a Snf2-class ATPase motor (*Clapier et al., 2017*; *Jungblut et al., 2020*; *Sundaramoorthy and Owen-Hughes, 2020*). Nucleosome remodeling is the result of ATP-dependent DNA translocation by the ATPase, which is locked onto the nucleosome through binding additional DNA- and histone sites. ATPase activity is directed by accessory subunits that determine targeting to specific genomic loci, and the outcome of the remodeling reaction. Different classes of remodelers, defined by their ATPase and unique sets of accessory subunits, are dedicated to distinct chromatin regulatory functions.

Members of the SWI/SNF family of remodelers have mainly been implicated in generating open chromatin at gene regulatory DNA elements (*Becker and Workman, 2013*; *Bracken et al., 2019*; *Brahma and Henikoff, 2020*; *Cakiroglu et al., 2019*; *Clapier et al., 2017*; *Pillidge and Bray, 2019*). Recent structural studies revealed that SWI/SNF-class remodelers engage the nucleosome in a highly modular manner (*Han et al., 2020*; *He et al., 2020*; *Jungblut et al., 2020*; *Mashtalir et al., 2020*; *Patel et al., 2019*; *Sundaramoorthy and Owen-Hughes, 2020*; *Wagner et al., 2020*; *Ye et al.,*

*2019*). The motor domain of the ATPase binds the nucleosomal DNA at superhelical position +2, while other modules contact the opposite DNA gyre, both faces of the histone octamer, and possibly the exiting DNA, and the histone H4 tail. SWI/SNF remodelers do not closely encompass the nucleosome, and the nucleosome remains mostly accessible. ATPase activity is only required for binding to a subset of genomic loci, but loss of ATPase activity can have dominant effects on chromatin binding, enhancer accessibility, and gene expression (*Gelbart et al., 2005*; *Hodges et al., 2018*; *Pan et al., 2019*). Mutations in key functional domains of SWI/SNF are associated with human cancer and developmental disorders (*Bracken et al., 2019*; *Cenik and Shilatifard, 2021*; *Mashtalir et al., 2020*; *Sundaramoorthy and Owen-Hughes, 2020*).

There are two main subclasses of SWI/SNF complexes that are conserved from yeast to man. The first includes yeast SWI/SNF, *Drosophila* BAP and mammalian BAF, while the second class includes yeast RSC, fly PBAP, and mammalian PBAF (*Bracken et al., 2019*; *Clapier et al., 2017*; *Mohrmann and Verrijzer, 2005*). Complexes of both classes share a common core, comprising related or identical subunits, associated with a set of signature subunits that define either SWI/SNF-BAF or RSC-PBAF. In mammalian cells, a third type of SWI/SNF complex has been described, named GBAF or non-canonical BAF (ncBAF; *Alpsoy and Dykhuizen, 2018*; *Gatchalian et al., 2018*; *Mashtalir et al., 2018*). *Drosophila* BAP and PBAP share a single ATPase, BRM, which is the ortholog of yeast Swi2/Snf2 and Sth1, and human SMARCA2 and SMARCA4. BRM associates with seven additional subunits to form the core of both BAP and PBAP. OSA (human ARID1A/B), D4/TTH (DPF1-3) and SS18 associate with the core subunits to form BAP, whereas POLYBROMO (PBRM1), BRD7, BAP170 (ARID2), and SAYP (PHF10) are specific for PBAF. The signature subunits play a major role in genomic targeting and functional selectivity of BAP and PBAP (*Bracken et al., 2019*; *Chalkley et al., 2008*; *Mohrmann et al., 2004*; *Moshkin et al., 2007*; *Moshkin et al., 2012*). Thus, SWI/SNF remodelers can be considered holoenzymes, in which different modules provide different functionalities that direct the remodeling activity of the ATPase. Although BAP and PBAP have shared activities, for example, both antagonize Polycomb repression, they also function in unique gene expression programs that control development, cell proliferation and differentiation (*Chalkley et al., 2008*; *Mohrmann et al., 2004*; *Moshkin et al., 2007*; *Moshkin et al., 2012*). Here, we use (P)BAP when making general statements that apply to both BAP and PBAP.

The natural amplification of *Drosophila* larval salivary gland polytene chromosomes allows the visualization of fluorescent-tagged transcription factors on interphase chromatin at native genetic loci in living cells. This has yielded fundamental insights in RNA polymerase II (RNAPII) recruitment and transcriptional dynamics (*Yao et al., 2007*; *Zobeck et al., 2010*). Salivary gland polytene chromosomes are the result of typically 10 rounds of DNA replication without segregation, resulting in cable-like super chromosomes comprising up to 1024 closely aligned chromatids. Interphase polytene chromosomes are visible by light microscopy and provided the first view of a physical genetic map (*Bridges, 1935*). The characteristic banding pattern of polytene chromosomes reflects the degree of interphase condensation, i.e., the ratio between the length of a stretched DNA molecule and the length of the corresponding chromosomal domain. The compacted polytene bands have been linked to transcriptionally repressed topologically associated domains (TADs), which are conserved in diploid cells (*Eagen et al., 2015*). Interbands contain gene regulatory regions, origins of replication, and are characterized by marks of active, open chromatin (*Zykova et al., 2018*). Polytene bands can be divided into moderately condensed gray chromatin, harboring the coding regions of active genes, and the highly compacted black chromatin that contains inactive developmental genes, and under-replicated, gene-poor intercalary heterochromatin (*Eagen et al., 2015*; *Zykova et al., 2018*). Very high levels of transcriptional activity results in puffing, the local uncoiling of individual chromosome strands due to the accumulation of the gene expression machinery and RNA. Pertinently, there is a close correspondence between chromatin structure of polytene chromosomes and that in diploid cells (*Eagen et al., 2015*; *Zykova et al., 2018*).

The steroid hormone 20-hydroxyecdysone (Ec) is the key regulator of *Drosophila* development and controls major transitions such as molting and metamorphosis (*Hill et al., 2013*). Exposure of larval salivary glands to Ec leads to the formation of early chromosomal puffs, starting a cascade of developmental gene expression and puffing. The Ec hormone mediates gene activation via binding to the Ec receptor (EcR), which associates with specific DNA elements. EcR belongs to the nuclear receptor family of transcription factors that mediate hormone-driven gene expression. Upon

hormone binding, nuclear receptors cooperate with a slew of coactivators, including the SWI/SNF remodelers, to stimulate transcription (*Hoffman et al., 2018*; *Paakinaho et al., 2017*).

Previous live-cell microscopy of fluorescent-tagged remodelers in diploid cells suggested a fast exchange dynamic, similar to many transcription factors (*Erdel et al., 2010*; *Johnson et al., 2008*; *Mehta et al., 2018*; *Paakinaho et al., 2017*; *Swinstead et al., 2016*; *Voss and Hager, 2014*). However, the in vivo kinetics of remodeler interactions with specific natural genomic loci has not been studied. In particular, the role of ATP hydrolysis in remodeler kinetics remains unclear. Here, we used vital imaging to determine the spatial and temporal dynamics of (P)BAP interactions with endogenous target loci and the effect of chromatin condensation. Our results show that ATP-hydrolysis fuels a continuous probing of the genome by (P)BAP. In the absence of ATP-hydrolysis, there is increased chromatin retention and reduced turnover of (P)BAP. We discuss the implications of our in vivo findings for understanding remodeler function and chromatin dynamics.

## Results

### The majority of BRM is not associated with chromatin

To visualize (P)BAP interactions with genomic loci in interphase cells, we imaged BRM in 3rd instar larval salivary gland polytene nuclei. In agreement with early studies (*Armstrong et al., 2002*; *Mohrmann et al., 2004*), visualization of endogenous BRM on fixated polytene chromosome spreads by epi-immunofluorescence (IF) revealed binding to loci within interbands, and to puffs (*Figure 1A*). BRM is largely excluded from heterochromatic bands, which stain strongly with DAPI and with antibodies directed against the core histones (α-HIS). Depletion of BRM affects does not lead to gross changes in the polytene chromosome banding pattern or the binding of RNAPII (*Figure 1—figure supplement 1A*). Whereas confocal IF imaging of BRM in formaldehyde-fixed whole mount salivary gland nuclei confirmed its association with chromosomal interbands, it also revealed that the majority of BRM resides in the nucleoplasm (*Figure 1B*). Indeed, co-localization analysis of BRM and DAPI yielded a Manders coefficient of 0.05, implying that only a small proportion of BRM is chromatin engaged at any given moment. A line-scan illustrates the preferential association of BRM with decondensed chromatin and its exclusion from highly compacted bands (*Figure 1C*). For comparison, the majority of RNAPII associated with well-defined chromosomal loci and has a lower abundance in the nucleoplasm (*Figure 1D*). In conclusion, our IF results show that BRM preferably interacts with interbands, but that, surprisingly, the majority (~95%) of BRM is not bound to chromatin.

To study the dynamics of BRM's interactions with polytene chromosomes, we generated transgenic *Drosophila* lines expressing fluorescent protein (either enhanced green fluorescent protein, GFP, or monomeric cherry, mCh)-tagged BRM, histone H2A and H2B. *Figure 1E* illustrates the expression of GFP-BRM in 3rd instar larval salivary gland nuclei. Laser scanning confocal microscopy of GFP-H2A and mCh-H2B in isolated salivary glands revealed strong co-localization, and recapitulation of the polytene banding pattern (*Figure 1F*). Thus, local chromatin condensation can be captured by live cell imaging of cultured salivary glands. Indeed, the intensity of the GFP-H2B signal corresponded to the highly compacted chromocenter, telomers and heterochromatic bands (black chromatin; yellow arrowhead), gray bands (red arrowhead) and white chromatin interbands (blue arrowhead; *Figure 1G*). Previous analysis of a range of chromatin marks suggested that approximately 5% of *Drosophila* genomic DNA comprises promoters and regulatory sequences, forming white chromatin. About 25% of the genome corresponds to gray chromatin, harboring active genes, whereas the remaining 70% represents black heterochromatin, comprising mainly intercalary heterochromatin and inactive genes (*Eagen et al., 2015*; *Filion et al., 2010*; *Szabo et al., 2019*; *Kharchenko et al., 2011*; *Zykova et al., 2018*). To determine the level of chromosomal condensation, we measured the intensity of GFP-H2B across multiple nuclei. First, we quantified the GFP-H2B intensity distribution across z-stacks of multiple individual nuclei. Next, we binned GFP-H2B intensities in three classes, corresponding to either white, gray, or black chromatin (*Figure 1—figure supplement 1B*). This analysis indicated that the volume of polytene chromosomes in nuclear space comprises about 5 (±1) % black, 43 (±4) % gray, and 52 (±5) % white chromatin (*Figure 1H*). Applying these estimates of GFP-H2B density distribution in living cells implies that chromatin in gray bands is on average sixfold more condensed than in interbands, whereas black chromatin is condensed an

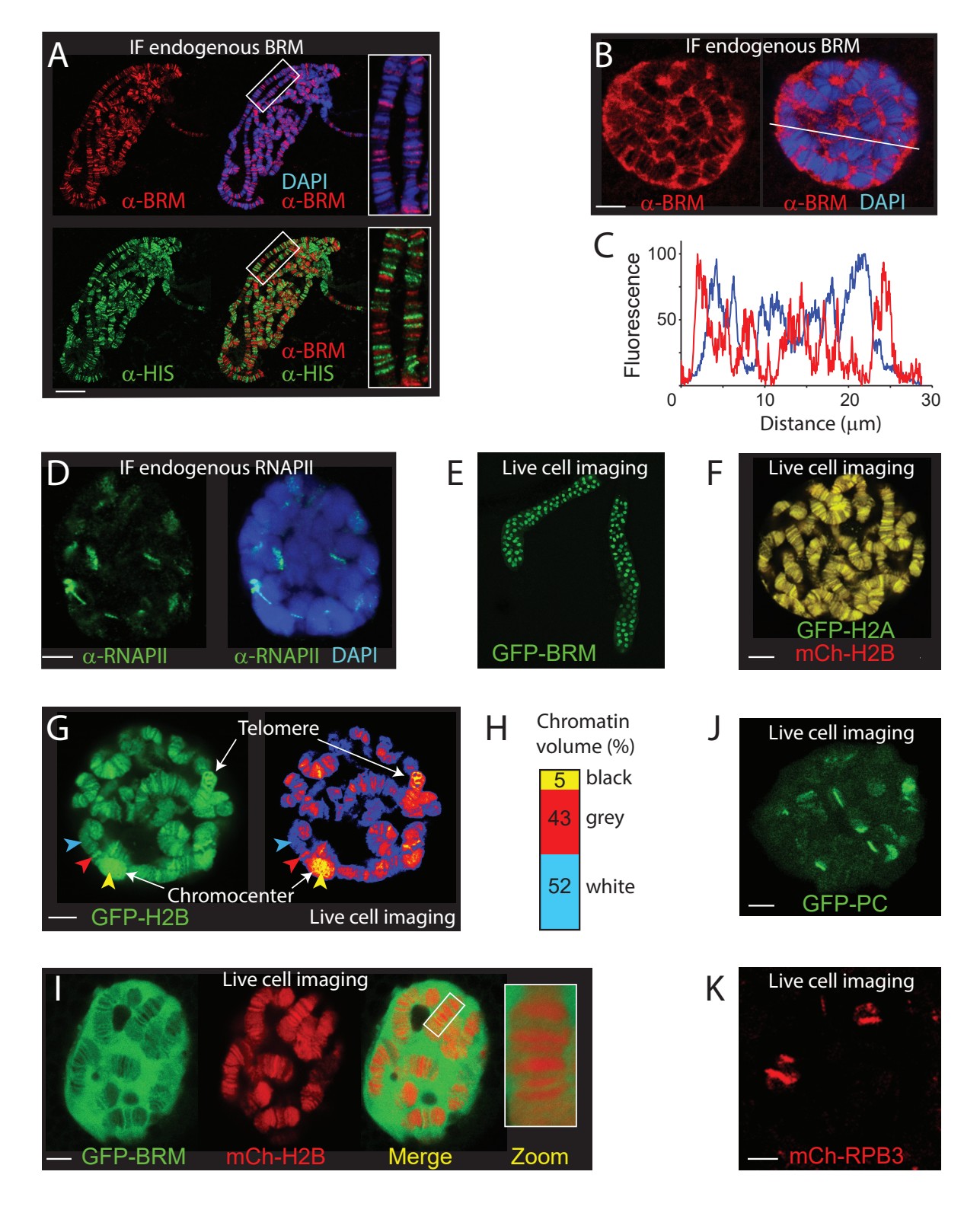

**Figure 1.** The chromosomal and nuclear distribution of BRM in larval salivary gland polytene nuclei. (**A**) Distribution of endogenous BRM on a polytene chromosome spread determined by indirect IF using antibodies against BRM (red) and core histones (HIS, green). DNA was visualized by DAPI staining (blue). Scale bar represents 20 µm. (**B**) Confocal IF image of whole mount intact polytene nucleus from a formaldehyde-fixed salivary gland using antibodies against BRM (red) and DAPI (blue). White line indicates the line scan used for the fluorescence intensity plot shown in panel C. Scale bar

*Figure 1 continued on next page*

*Figure 1 continued*

represents 5 µm. (C) Fluorescence intensity plot of BRM (red) and DNA (blue) across an intact polytene nucleus. Position of the line scan (from left to right) is indicated in (B). Fluorescence intensity is expressed as percentage of the highest pixel intensity of the entire image. Images in panel A-C are representative for polytene nuclei in salivary glands obtained from multiple larvae. (D) IF confocal section of whole mount intact polytene nucleus from a formaldehyde-fixed salivary gland using antibodies against RNAPII subunit RPB1 (red) and DAPI (blue). Scale bar represents 5 µm. (E) Image of an isolated, cultured 3$^{rd}$ instar larval salivary gland expressing GFP-BRM. (F) Confocal image of GFP-H2A and mCh-H2B in a cultured salivary gland reveals strong co-localization and recapitulates the polytene banding pattern. Scale bar represents 5 µm. (G) Confocal section of a GFP-H2B expressing polytene nucleus. The right panel provides a confocal false color image heat map of GFP fluorescence. Arrowheads indicate examples of white- (blue arrowhead), gray- (red arrowhead), or black- (yellow arrowhead) chromatin. The heterochromatic chromocenter and a telomeric region are indicated. Scale bar represents 5 µm. (H) The intensity of GFP-H2B was measured across 10 nuclei, nine confocal slices per nucleus. For raw data see *Figure 1— source data 1*. Next, GFP-H2B intensities were binned in three classes, corresponding to white, gray, and black chromatin, using thresholds obtained from the fluorescence intensity curves shown in *Figure 1—figure supplement 1B*. The resulting chromosome volumes of white, gray, and black chromatin are presented as a bar graph. (I) Confocal section of a polytene nucleus expressing GFP-BRM and mCh-H2B. (J) Confocal image of GFP-PC and (K) mCh-RPB3. Images in panels I-J are representative for polytene nuclei in salivary glands obtained from multiple independent crosses. Scale bars represents 5 µm.

The online version of this article includes the following source data and figure supplement(s) for figure 1:

**Source data 1.** GFP-H2B intensity across nuclei.
**Figure supplement 1.** Expression, incorporation in (P)BAP and localization of GFP-BRM and GFP-SNR1 in 3rd instar salivary gland polytene nuclei.
**Figure supplement 1—source data 1.** Original blots.

additional ~25-fold (up to ~150 more than open chromatin). Thus, visualization of fluorophore-tagged histones can be used to monitor chromatin condensation in living cells.

Next, we determined the distribution of GFP-BRM in live polytene nuclei. MOR (SMARCC1/2) co-immunopurifications (co-IPs) from dissected larval salivary gland extracts revealed the efficient incorporation of GFP-BRM in remodeler complexes (*Figure 1—figure supplement 1C*). IF of polytene chromosomes showed extensive co-localization of GFP-BRM with endogenous MOR, confirming that it binds chromatin normally as part of (P)BAP (*Figure 1—figure supplement 1D*). MOR has a key architectural function that is essential for the structural integrity of (P)BAP and the stability of BRM in vivo (*Moshkin et al., 2007*). RNAi-mediated depletion of MOR in the larval salivary gland caused a concomitant loss of GFP-BRM, indicating that it predominantly exists as part of (P)BAP (*Figure 1— figure supplement 1E*). We conclude that GFP-BRM is incorporated in (P)BAP and is targeted correctly. Vital imaging of salivary gland nuclei expressing mCh-H2B and GFP-BRM showed that the majority of GFP-BRM is nucleoplasmic and excluded from the nucleolus (*Figure 1I*). GFP-BRM associated with interband chromatin, while its level was strongly reduced within chromosome bands. We made similar observations for another (P)BAP core subunit, GFP-SNR1 (SMARCB1; *Figure 1—figure supplement 1F*). Pertinently, BRM and SNR1 levels within the condensed polytene chromosome bands are much lower than in the nucleoplasm. Note that loss of BRM does not affect polytene chromosome condensation (*Figure 1—figure supplement 1A*). These observations raise the intriguing possibility that chromosome condensation leads to (P)BAP exclusion. In contrast to BRM, GFP-Polycomb (PC) and the RNAPII subunit mCh-RBP3 each showed strong binding to specific chromosomal loci and relatively low abundance in the nucleoplasm (*Figure 1J and K*). In conclusion, a surprisingly small proportion of (P)BAP engages the genome at a steady state level. (P)BAP interacts with open chromatin but is largely excluded from highly condensed chromosome bands.

## PBAP, but not BAP, is required for Ec-induced gene expression

We were interested in (P)BAP dynamics during developmental gene activation. Therefore, we investigated the role of (P)BAP in Ec-controlled gene transcription. We dissected salivary glands from early 3rd instar larvae and cultured them for ~1 hr, either in the presence or absence of Ec. IF of fixated polytene chromosome spreads revealed binding of the EcR to the *E74* and *E75* loci, harboring Ec early response genes (*Figure 2A*). Like the EcR, BRM is present on the *E74* and *E75* loci both before and after induction. Concomitant with RNAPII recruitment and puffing (*Figure 2B*), the amount of BRM at *E74* and *E75* increased substantially following Ec-induction. Next, we compared the recruitment of the BAP-specific subunit OSA (ARID1A/B) and the PBAP-specific PBRM (*Figure 2C*). Before induction, both OSA and PBRM were present at *E74* and *E75*. After puffing, PBRM accumulated on the *E74* and *E75* loci, but we detected no appreciable binding of OSA (*Figure 2D*). On the *E74* and

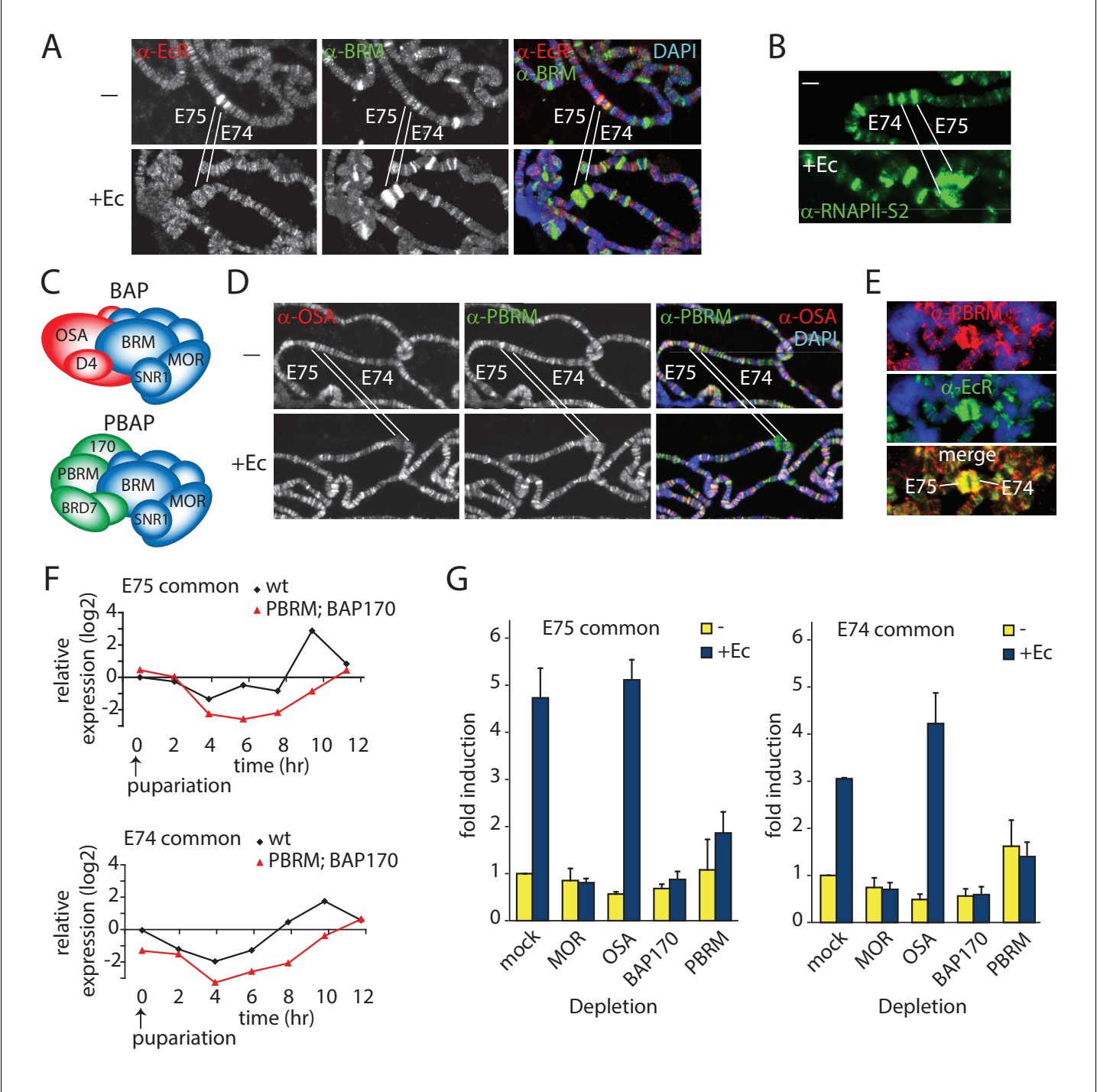

**Figure 2.** PBAP but not BAP is required for Ec-induction of the *E74* and *E75* loci. (**A**) Endogenous BRM and EcR binding to the *E74* and *E75* loci on 3rd instar larval polytene chromosomes before (-) and after activation by Ec, determined by indirect IF using antibodies against BRM (green) and EcR (red). DNA was visualized by DAPI staining (blue). The *E74* and *E75* loci are indicated. (**B**) IF of endogenous RNAPII recruitment using an antibody against RNAPII phosphorylated at Ser2 (RNAPII-S2). (**C**) Schematic representation of *Drosophila* BAP and PBAP. (**D**) Although both were present prior to induction, only PBAP but not BAP accumulates on the puffed *E74* and *E75* loci after activation by Ec. IF of endogenous OSA (red) or PBRM (green) before (-) and after addition of Ec. (**E**) IF of PBRM (red) and EcR (green) on the puffed *E74* and *E75* loci. (**F**) PBAP mutant larvae mis-express *E74* and *E75* transcripts. Wt and homozygous *Bap170^{kim1}; Pbrm^{33.2}* mutant prepupae were isolated at 2 hr intervals from pupariation (t=0) for 12 hr. RNA was extracted, and relative expression levels of E74 and E75 transcripts were determined by (RT)-qPCR. Shown are the results for primers corresponding to exons shared by distinct transcripts from *E74* and *E75*, respectively. Biological triplicates were used. Variance per time point was below 10%. (**G**) PBAP is required for the induction of *E74* and *E75* expression by Ec in S2 cells. RT-qPCR analysis of *E74* and *E75* expression in S2 cell in the absence (yellow)

*Figure 2 continued on next page*

*Figure 2 continued*

or presence of Ec (blue). Cells were treated with dsRNA directed against GFP (mock), MOR, OSA, BAP170, or PBRM. The relative level of expression in mock-treated cells in the absence of Ec was set at 1.

*E75* loci, PBRM co-localized with the EcR (*Figure 2E*). Thus, both BAP and PBAP bind the repressed *E74* and *E75* loci, but only PBAP is recruited after transcriptional activation by Ec. This observation agrees well with our earlier report that PBAP-specific mutants, but not OSA mutants, display micro-cephaly and leg malformations that are reminiscent of phenotypes caused by defective Ec-signaling (*Chalkley et al., 2008*). Collectively, these results suggest a role for PBAP in gene regulation by Ec. To test this notion, we compared the expression profiles of *E74* and *E75* transcripts in homozygous *Bap170*$^{kim1}$; *Pbrm*$^{33.2}$ prepupae to that of *wt* animals. Prepupae were collected at pupariation (t=0), and RNA was extracted at 2 hr intervals for 12 hr, and monitored by reverse transcription (RT)-qPCR (*Figure 2F*). Loss of BAP170 (ARID2) and PBRM disrupted the expression of the *E74* and *E75* genes, showing that PBAP is required for Ec-controlled gene regulation in vivo. Finally, we compared the requirement of MOR, OSA, BAP170 and PBRM for induction of *E74* and *E75* expression by Ec in S2 cells (*Figure 2G*). Again, the loss of PBAP signature subunits, but not OSA, abrogated gene induction by Ec. Collectively, these results show that PBAP is required for Ec-induced developmental gene expression.

## Chromatin condensation at Ec-induced puffs

To study Ec-regulated loci on polytene chromosomes in live salivary glands, we generated transgenic *Drosophila* lines expressing GFP- or mCh-tagged EcR. Confocal imaging of GFP-EcR and mCh-H2B allowed us to readily identify the *E74* and *E75* gene clusters in cultured salivary gland nuclei (*Figure 3A*). Co-expression of mCh-EcR with GFP-BRM illustrates the highly defined genomic locali-zation of the sequence-specific transcription factor compared to the diffuse distribution of the remodeler (*Figure 3B*). A high-resolution zoom shows the puffing of the chromatin bound by EcR, which are flanked by condensed bands (*Figure 3C*). The EcR binding sites within the *E74* and *E75* loci have been mapped by ChIP and cover approximately 40 and 60 kb, respectively (*Bernardo et al., 2014*). We plotted the EcR sites on the genomic map, and indicated the simplified local chromatin state (white, gray, or black), which was derived from combining available maps of histone marks (*Filion et al., 2010*; *Kharchenko et al., 2011*; *Figure 3D*). Two well-defined hetero-chromatic bands (fragments I and III) flank the GFP-EcR-marked *E75* puff (fragment II; *Figure 3C and D*). We measured the lengths of these fragments in 20 different nuclei, allowing us to compare observed physical distances in living cells with genomic DNA lengths (*Figure 3E*). The packing ratio of an 11 nm nucleosomal array is predicted to be 6.8:1. This would yield an extended fiber length for fragment I of ~18 µm. However, the observed average length of fragment I in living polytene nuclei is only 0.72 (± 0.18) µm, yielding a 25-fold compaction. Similarly, the heterochromatic band (fragment III) flanking E75 appears to be condensed ~22-fold, compared to an extended 11 nm chro-matin fiber. In contrast, measurement of the GFP-EcR marked puff (fragment II) suggested a com-paction of only threefold. Given that the EcR-bound enhancers and promoters form loops (*Bernardo et al., 2014*), and that cellular chromatin is a flexible disordered polymer, these observa-tions are consistent with the notion that the puffed area comprises an open array of nucleosomes. When we considered the polytene chromosome as a cylinder, and included diameter measurements to calculate apparent volumes, the difference in chromatin compaction between puffs and bands became even more pronounced. For example, the chromatin in the heterochromatic fragment I appears to be ~30-fold more compacted than that in the GFP-EcR-marked puff of fragment II. Thus, vital imaging of fluorophore-tagged histones on polytene chromosomes allows measurements of rel-ative interphase chromatin condensation at specific loci in living cells.

## Loss of ATP-binding enhances BRM chromatin retention and clustering in S2 cells

Our analysis of (P)BAP in polytene nuclei showed that only a fraction of (P)BAP engages chromatin at a given moment. This raises the question of remodeler turnover on chromatin and the role of ATP-hydrolysis in this process. To address this issue, we first generated S2 cell lines that express

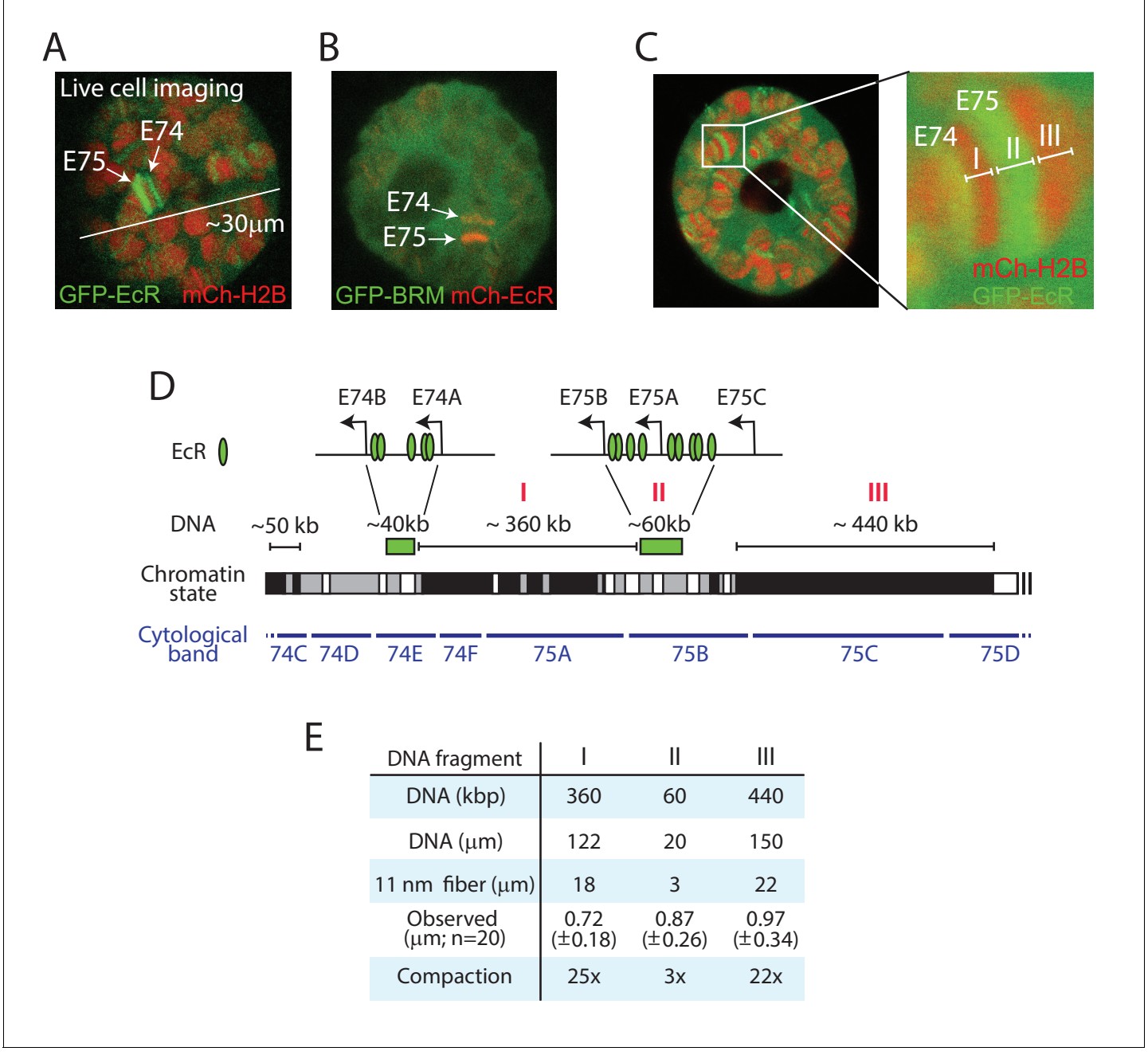

**Figure 3.** Live imaging reveals local chromatin condensation at the *E74* and *E75* loci. (**A**) Confocal image of a polytene nucleus expressing GFP-EcR and mCh-H2B. *E74* and *E75* are indicated. (**B**) Confocal image of a nucleus expressing GFP-BRM and mCh-EcR. (**C**) Confocal image and zoom of the *E74* and *E75* loci in a nucleus expressing GFP-EcR and mCh-H2B. The GFP-EcR-marked *E75* puff (fragment II) flanked by two heterochromatic bands (Fragments I and III) are indicated. (**D**) Chromosomal map of the E74 and E75 loci. The EcR binding sites (***Bernardo et al., 2014***) are indicated. The simplified local chromatin state (white, gray, or black) was derived from combining available maps of histone marks (***Filion et al., 2010***; ***Kharchenko et al., 2011***). The GFP-EcR-marked *E75* puff (fragment II) and flanking heterochromatic bands (fragments I and III) are indicated with red roman numerals. (**E**) Comparison between observed chromosomal physical distances in living cells and genomic DNA length. In vivo apparent lengths of fragments I, II and III (see panels C and D) were determined in 20 different nuclei in cultured salivary glands. The degree of in vivo compaction was derived by comparing observed fragment length to the calculated length of the corresponding DNA packaged into an extended 11 nm nucleosomal fiber (packing ratio of 6.8:1).

either GFP-BRM or a catalytically inactive ATP-binding mutant, GFP-BRM-K804R (*Elfring et al.,*
*1998*). Western blotting established that GFP-BRM and GFP-BRM-K804R were expressed at comparable levels, which were well below that of endogenous BRM (*Figure 4A*). IPs with antibodies directed against GFP followed by mass spectrometry showed that both GFP-BRM and GFP-BRM-K804R associated with the full complement of BRM complex subunits (*Figure 4B*). Interestingly, we identified a homolog of the mammalian GBAF-specific subunit GLTSCR1/L in the GFP-BRM IPs, suggesting that *Drosophila* contains a corresponding complex. Both GFP-BRM and GFP-BRM-K804R

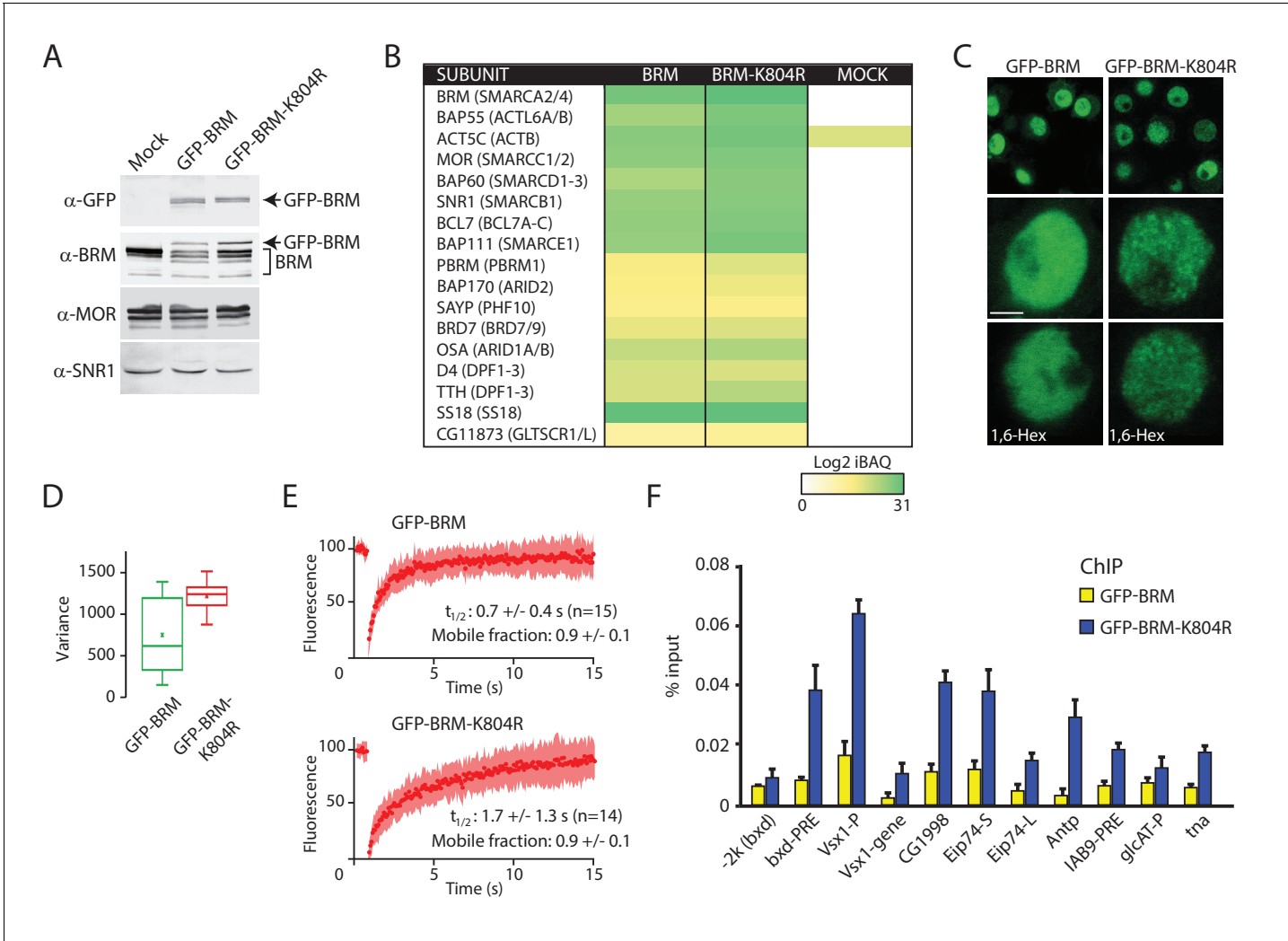

**Figure 4.** Enhanced clustering and chromatin retention of a BRM ATP-binding mutant in S2 cells. (**A**) Western blot analysis of extracts from *Drosophila* S2 cells that were either transfected with empty vector or with a vector expressing GFP-BRM or GFP-BRM-K804R. Blots were probed with the indicated antibodies. For original blots see *Figure 4—source data 1*. (**B**) Mass spectrometric analysis of GFP-BRM-associated proteins immunopurified using antibodies directed against GFP. *Drosophila* (P)BAP subunits and their human orthologs (in brackets) are indicated. The color gradient reflects the Log2 iBAQ scores. In addition to all BAP and PBAP subunits (*Bracken et al., 2019*), we also detected the *Drosophila* homolog of human GLTSCR1. Note that fly BRD7 is homologous to both human BRD7 and BRD9. (**C**) Confocal images of GFP-BRM or GFP-BRM-K804R expressing S2 cells. Bottom panels have been treated with 1,6-Hex at a final concentration of 5% for 1 min. Scale bar represents 1 μm. (**D**) Box and Whisker plots of the mean variance of the nuclei after applying a variance filter (ImageJ, 2-pixel radius) to the GFP fluorescence images of GFP-BRM and GFP-BRM-K804. Mean +/- SD; 880 +/- 432 (n=29) for GFP-BRM and 1351 +/- 171 (n=32) for GFP-BRM-K804R, p-value: $2.7 \times 10^{-6}$ (Student T test). Units variance: pixel intensity$^2$. (**E**) FRAP of GFP-BRM and GFP-BRM-K804R after bleaching a 16-pixel wide strip across the nucleus. Curves represent the average +/- SD of 15 (GFP-BRM) or 14 (GFP-BRM-K804R) independent experiments. (**F**) ChIP-qPCR analysis of GFP-BRM (yellow) or GFP-BRM-K804R (blue) binding to a subset of functional (P) BAP target loci (*Moshkin et al., 2012*). The −2 k (*bxd*) sequence is a negative control.

The online version of this article includes the following source data for figure 4:

**Source data 1.** Original blots.

were nuclear and excluded from the nucleolus (*Figure 4C*). Although they were expressed at comparable levels, GFP-BRM and GFP-BRM-K804R displayed notably different patterns of distribution. Although wild type (wt) GFP-BRM was distributed evenly throughout the nucleoplasm, GFP-BRM-K804R displayed a punctate pattern. Quantification of local variance of GFP intensity confirmed the clustering of GFP-BRM-K804R (*Figure 4D*). Recently, liquid-liquid phase separation driven by weak hydrophobic interactions has been implicated in localized enrichment and condensation of proteins (*McSwiggen et al., 2019*; *Plys and Kingston, 2018*). Poly-alcohols, such as 1,6-hexanediol (1,6-Hex) that disrupt weak hydrophobic interactions, have become popular as a diagnostic tool to identify phase separation in biological processes. However, the addition of 1,6-hexanediol did not affect the punctate pattern of GFP-BRM-K804R, indicating that phase separation does not explain the clustering of GFP-BRM-K804R (*Figure 4C*).

To gain insight in the dynamic behavior of GFP-BRM and GFP-BRM-K804R, we compared their fluorescence recovery after photobleaching (FRAP) profiles after bleaching a 16-pixel wide strip across the center of the nucleus (*Figure 4E*). The turnover of GFP-BRM-K804R was approximately two-times slower than that of GFP-BRM, with a time of half recovery ($t_{1/2}$) of ~ 1.7 (± 1.3) seconds (s) compared to ~ 0.7 (± 0.4) s for wt BRM. Cancer-associated mutations in the ATP-binding cleft of human SMARCA4 caused a comparable ~ 2-fold reduction in FRAP recovery kinetics in diploid cells (*Hodges et al., 2018*). These observations suggest a role for ATP binding and hydrolysis in remodeler dynamics. The recovery curves did not identify a substantial immobile fraction for either GFP-BRM-K804R or GFP-BRM. Next, we used chromatin immunoprecipitation quantitative PCR (ChIP-qPCR) assays to compare the binding of GFP-BRM and GFP-BRM-K804R to selective (P)BAP target loci that were identified earlier (*Moshkin et al., 2012*). The catalytically inactive BRM-K804R yielded two- to fivefold stronger ChIP signals at a variety of functional binding sites than wt BRM (*Figure 4F*). The increased chromatin binding of the BRM ATP-binding mutant to functional target loci is reminiscent of earlier observations for the yeast Iswi2 remodeler (*Gelbart et al., 2005*). Collectively, these results indicate that the loss of ATP-binding causes chromatin retention and clustering of BRM-K804R.

## (P)BAP interacts transiently with chromatin

Vital imaging of BRM in S2 cells revealed a fast exchange dynamic, suggestive of transient probing of the genome. However, it is impossible to visualize transcription factor interactions with genomic loci in regular interphase cells. To overcome this limitation, we imaged BRM in salivary gland polytene nuclei. First, we used fluorescence loss in photobleaching (FLIP)-FRAP to determine the in vivo dynamics of GFP-BRM in salivary gland polytene nuclei. Following photobleaching of half a nucleus, we observed a rapid recovery that was largely complete after ~2 min (*Figure 5A and B*). The increase in fluorescence in the bleached part of the nucleus was accompanied by a comparable decrease in the unbleached part. We noted no difference between BRM in the nucleoplasm or associated with chromatin (*Figure 5A*). In contrast to GFP-BRM, mCh-H2B showed no recovery within a timeframe of minutes (min; *Figure 5C and D*). Thus, (P)BAP can move freely through the nucleus, whereas the turnover of bulk histones is limited. FRAP analysis of GFP-BRM bands on chromosomes also showed a rapid and complete recovery of fluorescence (*Figure 5E and F*). Kinetic analysis revealed a recovery $t_{1/2}$ of 3.1 (±1.1) s and no substantial immobile fraction. Similar FRAP kinetics were observed for a low-expressing GFP-BRM *Drosophila* line (*Figure 5—figure supplement 1A–C*). Moreover, FRAP analysis of GFP-SNR1 (SMARCB1) gave comparable results (*Figure 5G* and *Figure 5—figure supplement 1D–F*). Finally, co-localization with RNAPII did not affect the FRAP kinetics of GFP-BRM substantially (*Figure 5—figure supplement 1G,H*). The rapid exchange and absence of an immobile fraction shows that the interactions of (P)BAP with chromatin are highly dynamic, with retention times of just a few seconds. Consequently, (P)BAP kinetics are dominated by free diffusion in the nucleoplasm.

The RNAPII subunit GFP-RBP3 displayed a very different kinetic profile. In agreement with earlier studies (*Yao et al., 2007*), photobleaching of chromatin associated GFP-RPB3 resulted in an initial rapid increase in fluorescence, followed by a second, more prolonged linear phase of recovery which lasted for several min (*Figure 5G*). The rapid recovery at the start probably reflects dynamic binding to the promoter, whereas the linear phase corresponds to elongating RNAPII. The turnover of bulk histone H2B is more than two orders of magnitude slower than that of (P)BAP. FRAP analysis of GFP-H2B in polytene chromosomes revealed that it takes over 40 min to obtain a ~20% recovery

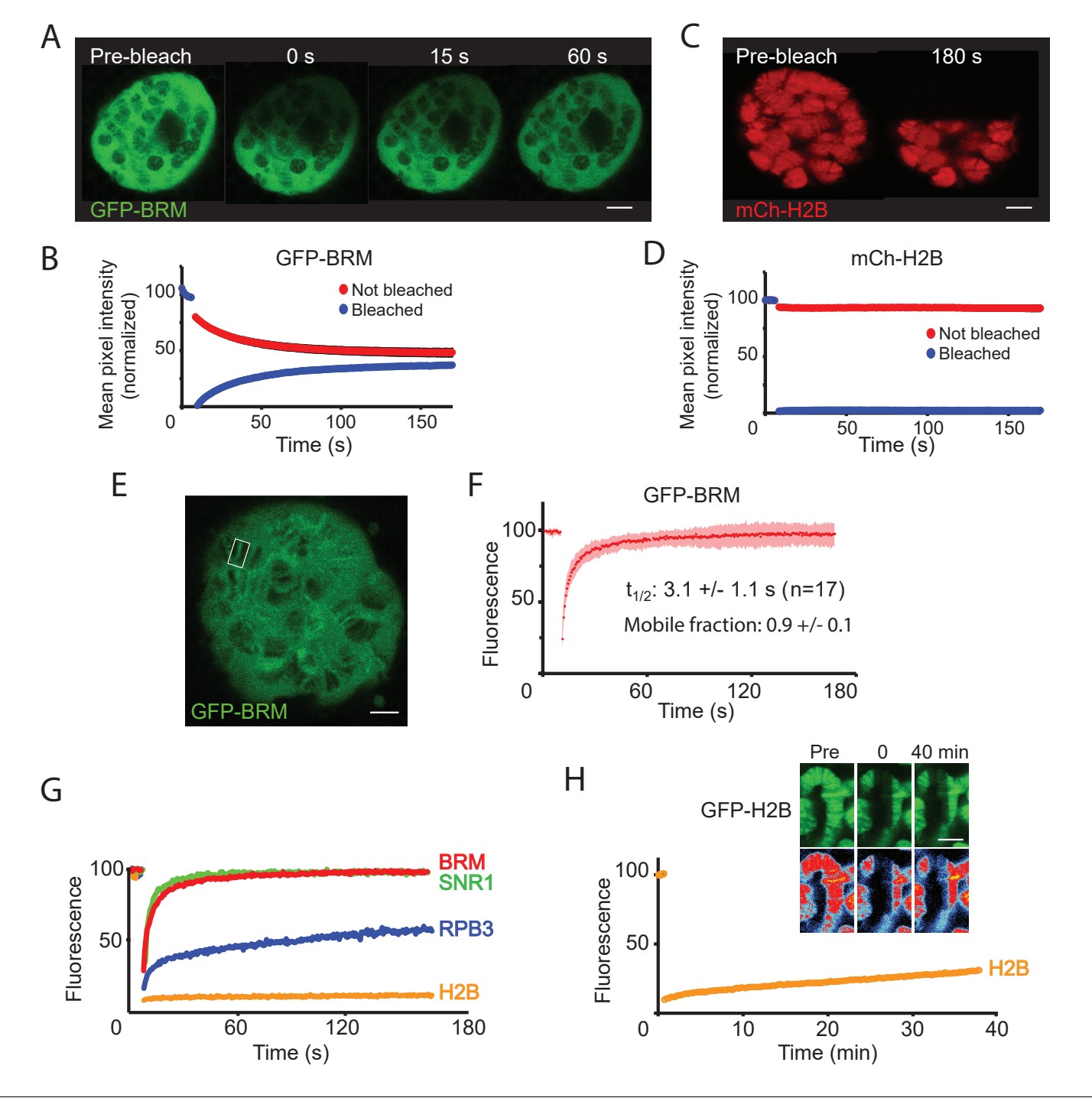

**Figure 5.** BRM interacts transiently with chromatin. (A) Confocal images of GFP-BRM expressing salivary gland nuclei before or 0, 15 or 60 s after photo-bleaching the upper half of the nucleus. (B) Corresponding average FLIP-FRAP curves of GFP-BRM (n=6; SD bleached curve < 2%; unbleached curve < 1%). (C) Confocal images of mCh-H2B expressing salivary gland nuclei before or 180 s after photo-bleaching the upper half of the nucleus. (D) Corresponding average FLIP-FRAP curves of mCh-H2B (n=9; SD bleached curve < 1%; unbleached curve < 3%). (E) Representative confocal image of a chromosomal band of GFP-BRM analyzed by FRAP. Rectangle indicates bleached area. (F) Averaged FRAP curves of chromosomal GFP-BRM. Data are expressed as mean ± SD for 17 nuclei. Half maximal recovery time ($t_{1/2}$) and size of the mobile fraction were determined as described in the Materials and methods section. (G) Chromosomal band FRAPs of the (P)BAP subunits GFP-BRM (red) and GFP-SNR1 (green), RNAPII subunit mCh-RPB3 (blue) and GFP-H2B (yellow). Traces shown are the average of 17 (GFP-BRM), 14 (GFP-SNR1), 9 (mCh-RPB3), and 18 (mCh-H2B) experiments. GFP-BRM data are obtained from *Figure 3F*. (H) Confocal images of a GFP-H2B containing chromosome segment before, directly after and 40 min after

*Figure 5 continued on next page*

*Figure 5 continued*

photobleaching. Top images: GFP-fluorescence; bottom images; heat map. Graph shows the fluorescence recovery of mCh-H2B. All scale bars represent 5 μm.

The online version of this article includes the following figure supplement(s) for figure 5:

**Figure supplement 1.** Localization and mobility of GFP-SNR1 and GFP-BRM in a low expressing *Drosophila* line.

(*Figure 5H*). GFP-H2B recovery in polytene bands versus interbands appeared to be proportional, consistent with the notion that both comprise the same basic nucleosomal fiber at different degrees of condensation (*Ou et al., 2017*).

To determine the turnover of BRM at highly transcribed loci that depend on BRM for expression, we activated *E74* and *E75* by adding Ec to isolated salivary glands. Ec induces prominent polytene chromosome puffs, which are a manifestation of very high levels of gene transcription (*Figure 6A,B*). The addition of Ec leads to increased RNAPII occupation of the *E74* and *E75* loci (monitored by GFP-RPB2), which peaks ~35 min after hormone addition (*Figure 6C,D* and *Figure 6—figure supplement 1A*). FRAP analysis of GFP-RPB2 at the peak of RNAPII accumulation (30 min), revealed only a partial (~60%) recovery that ceased after ~5 min (*Figure 6E*). The absence of a full recovery of RNAPII at the height of *E74* and *E75* expression and puffing might reflect either stalling or recycling. However, similar behavior of RNAPII on fully induced heat-shock loci was shown to be caused by local recycling (*Yao et al., 2007*). FRAP analysis of mCh-EcR revealed a recovery $t_{1/2}$ of 3.0 (± 1.1) s, which is comparable to BRM. In contrast to BRM, however, a small but consistent fraction of about 0.14 (± 0.07) of mCh-EcR appeared to be immobile (*Figure 6F*). Histones at the *E74* and *E75* puffs did not exchange notably (*Figure 6—figure supplement 1B*). Finally, we compared the targeting of the BAP-specific D4 (DPF1-3) subunit and PBAP-specific BRD7 by live cell imaging. In agreement with our results for endogenous BAP versus PBAP on fixated polytene spreads (*Figure 2D*), GFP-BRD7 co-localized with mCh-EcR on the *E74* and *E75* loci, whereas GFP-D4 did not (*Figure 6G*, *Figure 6—figure supplement 1C,D*). FRAP analysis of GFP-BRD7 revealed fast kinetics on the active *E74* and *E75* loci, which was comparable to that of BRM (*Figure 6H*). Likewise, GFP-D4 it displayed a fast turnover on BAP target loci (*Figure 6—figure supplement 1E*). In summary, our FRAP results suggest that at the peak of induction, a substantial portion of RNAPII is recycled at the *E74* and *E75* puffs. However, PBAP, which is essential for expression of *E74* and *E75*, exchanges rapidly without a substantial immobile fraction. We conclude that (P)BAP acts through a continuous and rapid probing of the genome. By comparison, the bulk turnover of histones is limited and slow.

## ATP-hydrolysis stimulates (P)BAP release from chromatin

To investigate the role of ATP hydrolysis in (P)BAP-chromatin interactions, we permeabilized salivary glands with digitonin, using a method developed to study Ec-induced puff formation (*Myohara and Okada, 1987*). Digitonin-treated glands are permeable to high-molecular-weight molecules but dependent upon the addition of ribonucleoside triphosphates, retain the ability to form puffs in response to Ec. Treatment of cultured salivary glands with 0.01% digitonin resulted in a complete loss of GFP fused to a nuclear localization signal (NLS-GFP), whereas mCh-H2B remained bound to the chromosomes (*Figure 7A*). In striking contrast, chromatin-binding of GFP-BRM increased dramatically following permeabilization. Moreover, free GFP-BRM was no longer detectable in the nucleoplasm. Conversely, chromatin association of GFP-PC and mCh-RBP3 was substantially reduced in permeabilized glands. Although BRM normally only interacts transiently with chromatin and is mainly nucleoplasmatic, heat maps of GFP-BRM illustrate that permeabilization resulted in strong accumulation of GFP-BRM onto chromatin (*Figure 7B*). Chromatin-bound GFP-BRM in permeabilized cells was essentially immobile, as determined by FRAP (*Figure 7C*). Following permeabilization, GFP-BRM remained bound for more than an hour. Tellingly, upon the addition of ATP (2 mM), GFP-BRM was immediately released from chromatin and diffused out of the nuclei (*Figure 7D*). We used this assay to assess the nucleotide requirements for GFP-BRM release (*Figure 7E*). First, ATP could not be replaced by ADP. Second, efficient release of GFP-BRM required ATP concentrations above 1 mM. Third, neither the slowly hydrolyzed analogue ATP-γ-S nor the non-hydrolysable ATP ground state mimetic ADP-AlF$_4^-$ could replace ATP. Thus, the release of GFP-BRM from polytene chromosomes requires hydrolysable ATP at concentrations greater than 1 mM. Note that ATP

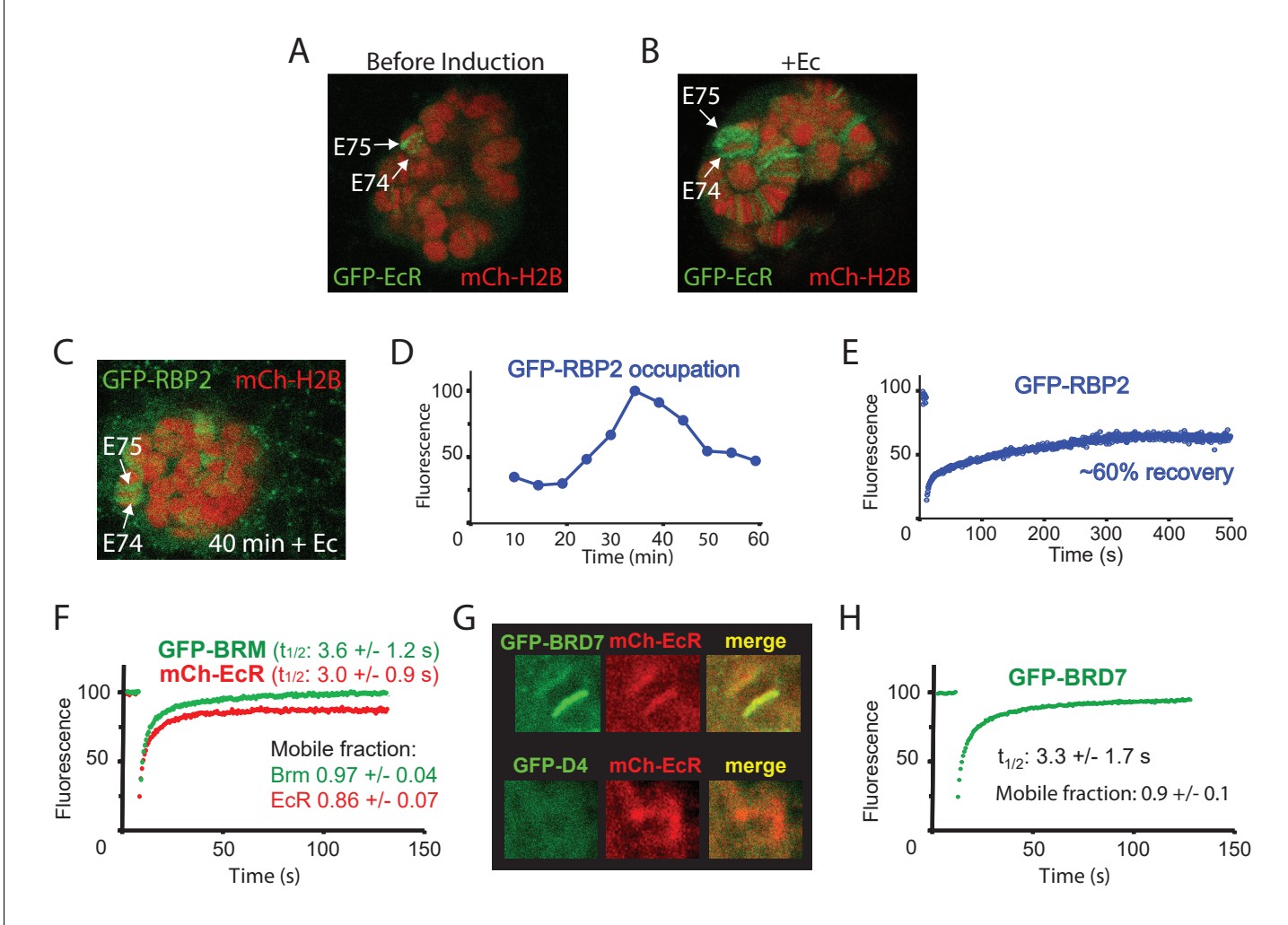

**Figure 6.** Dynamics of EcR, RNAPII and PBAP at the *E74* and *E75* loci. (**A**) Confocal image of a polytene nucleus expressing GFP-EcR and mCh-H2B prior to- and (**B**) 40 min following addition of Ec. (**C**) Confocal image of a salivary gland nucleus expressing GFP-RBP2 and mCh-EcR 40 min after the addition of Ec. (**D**) Addition of Ec to isolated salivary glands leads to increased RNAPII occupation of the *E74* and *E75* loci. GFP-RBP2 accumulation on the *E74* and *E75* loci after addition of Ec. Fluorescence is expressed as percentage of the maximal mean pixel intensity. See also *Figure 6—figure supplement 1A*. (**E**) FRAP analysis of GFP-RPB2 at the peak of RNAPII accumulation (30 min). (**F**) Averaged FRAP curves of GFP-BRM (green) and mCh-EcR (red) at the *E74* and *E75* loci. Data are expressed as mean ± SD for n=10 nuclei. (**G**) GFP-BRD7, but not GFP-D4, co-localizes with mCh-EcR on the *E74* and *E75* loci. Confocal images of nuclei expressing mCh-EcR (red) and either GFP-BRD7 (green) or GFP-D4 (green). See also *Figure 6—figure supplement 1C,D*. (**H**) Averaged FRAP curves of GFP-BRD7 (green) on the *E74* and *E75* loci.

The online version of this article includes the following figure supplement(s) for figure 6:

**Figure supplement 1.** Ec-induced transient recruitment of RNAPII to the E74 – E75 locus.

concentrations in living cells normally do not drop to a level below 1 mM, when it would impede remodeler dynamics. To monitor the effect on endogenous (P)BAP, we repeated the permeabilization in either the presence- or absence of ATP, followed by fixation and IF of MOR (SMARCC1/2). Similar to our vital imaging results, we found that MOR accumulated on chromatin in permeabilized glands but dissociated and diffused out of the nucleus upon addition of ATP (*Figure 7F*). Thus, endogenous (P)BAP and GFP-BRM show similar behavior. Taken together, these results show that ATP hydrolysis is required for (P)BAP release, but not for binding to chromatin.

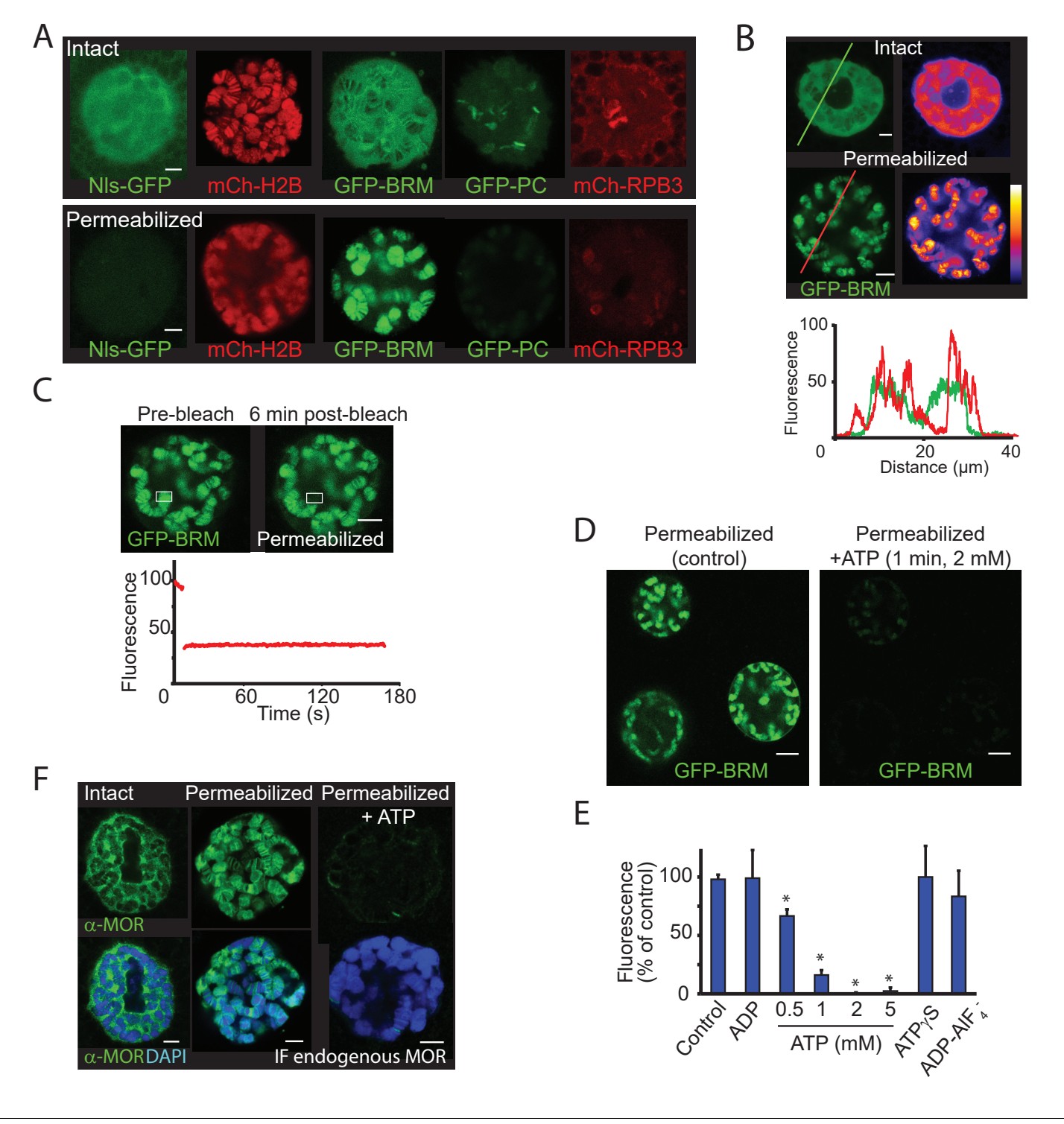

**Figure 7.** ATP-hydrolysis is required for (P)BAP release from polytene chromosomes. (**A**) Confocal images showing the nuclear distribution of NLS-GFP, mCh-H2B, GFP-BRM, GFP-PC, and mCh-RPB3 in isolated salivary glands that were either intact (top panels) or permeabilized by treatment with 0.01% digitonin (bottom panels). Images are representative for polytene nuclei in salivary glands obtained from multiple independent crosses. (**B**) GFP-fluorescence and heat maps of GFP-BRM in a nucleus of an intact or a digitonin-permeabilized salivary gland. Fluorescence intensity plots corresponding to the line scans (from left to right) indicated in the top panels. Fluorescence intensity is expressed as percentage of the highest pixel intensity of the combined images. The false color scale bar represents full-scale gray values (eight bit) from 0 (black) to 255 (white). (**C**) FRAP of a GFP-BRM chromosomal band in a nucleus of a digitonin-permeabilized salivary gland. Rectangle indicates the bleached area. (**D**) Confocal images of GFP-

*Figure 7 continued on next page*

Figure 7 continued

BRM in a permeabilized gland prior to- or 1 min after the addition of ATP to a final concentration of 2 mM. See also *Figure 7—video 1* depicting GFP-BRM after addition of ATP. (E) Effect of adenosine nucleotides and ATP mimetics on chromatin retention of GFP-BRM in permeabilized salivary glands. Fluorescence intensity (as percentage of control without added ATP) of individual nuclei in permeabilized salivary glands incubated for 5 min with 5 mM ADP, 0.5–5 mM ATP, 5 mM ATP-γ-S or ADP-AlF$_4^-$ (5 mM ADP + 10 mM AlF$_4^-$). Data are expressed as mean +/- SD. Asterisk indicates a significant difference from the control (p < 0.05). (F) Endogenous (P)BAP clamps chromatin in the absence of ATP. Confocal IF images of whole mount nuclei from intact or permeabilized salivary glands using antibodies against MOR (green) and DAPI (blue). Prior to fixation, permeabilized glands were incubated in media that either lacked- or contained 5 mM ATP for 10 min. Images are representative for polytene nuclei in salivary glands obtained from multiple larvae. All scale bars represent 5 μm.

The online version of this article includes the following video for figure 7:

**Figure 7—video 1.** Video depicting GFP-BRM in permeabilized salivary gland after addition of ATP.

https://elifesciences.org/articles/69424#fig7video1

## BRM-K804R traps wt BRM onto polytene chromosomes and impedes histone turnover

To complement the ATP-depletion experiments in permeabilized cells, we analyzed the dynamics of the ATP-binding deficient GFP-BRM-K804R in polytene salivary glands. BRM-K804 acts as a dominant negative mutant in developing *Drosophila*. Depending on its level of expression, BRM-K804 causes homeotic transformations or lethality (*Elfring et al., 1998*). IF of polytene chromosome spreads revealed co-localization of GFP-BRM-K804R with endogenous MOR (SMARCC1/2), indicating largely normal targeting of this mutant (*Figure 8A*). The chromosomes in GFP-BRM-K804R expressing glands, however, were considerably thinner than in wt glands, and had an increased tendency to break during preparation. These observations suggest that the expression of GFP-BRM-K804R leads to DNA under-replication and fragile polytene chromosomes. Vital imaging of intact salivary glands revealed that, in contrast to wt BRM, the majority of GFP-BRM-K804R is bound to chromatin (*Figure 8B*). Moreover, FRAP analysis showed that GFP-BRM-K804R has no substantial turnover (*Figure 8C*). Permeabilization with digitonin, either in the presence or absence of ATP, did not affect the chromosome-association of GFP-BRM-K804R (*Figure 8D and E*). IF of SNR1 (SMARCB1) showed that the expression of GFP-BRM-K804R forces binding of endogenous (P)BAP to polytene chromosomes (*Figure 8F*). The clustered binding of GFP-BRM-K804R prompted us to consider the potential role of phase separation. The addition of 1,6-hexanediol, however, affected neither the diffuse distribution of GFP-BRM nor the clustered binding to interband chromatin of GFP-BRM-K804R (*Figure 8G*). As reported (*Strom et al., 2017*), 1,6-hexanediol did reduce the accumulation of mCh-HP1 on the chromocenter. The insensitivity to 1,6-hexanediol and the lack of mobility revealed by FRAP suggest that phase separation does not play a major role in the clustering of GFP-BRM-K804R. Rather, we propose that inadequate release from chromatin drives the local accumulation of BRM-K804R (and that of wt BRM in the absence of ATP; see *Figure 7*). We note that the effect of the K804R mutation on BRM mobility is more pronounced in polytene nuclei than in S2 cells, which are near tetraploid. Most likely, the high local density of target loci in polytene chromosomes amplifies the effect of BRM-K804R retention on chromatin.

When co-expressed in the larval salivary glands, mCh-BRM-K804R altered the intra nuclear distribution of GFP-BRM dramatically. Instead of being ~95% nucleoplasmic, GFP-BRM now co-localizes with mCh-BRM-K804R on the chromosomes (compare *Figures 1I* and *8H*). Moreover, FRAP analysis showed that in the presence of mCh-BRM-K804R, GFP-BRM also became immobile (*Figure 8I*). Note that remodeler complexes contain a single ATPase subunit and do not multimerize in solution (*Jungblut et al., 2020*; *Sundaramoorthy and Owen-Hughes, 2020*). Nevertheless, BRM-K804R captures wt (P)BAP onto polytene chromosomes, possibly via a chromatin-mediated interaction. These results raise the intriguing possibility that remodeling involves direct cooperation, or a hand-off mechanism between different BRM complexes. Finally, we studied the impact of BRM-K804R on histone turnover. We used FRAP to determine histone exchange in salivary glands that co-expressed mCh-H2B with either GFP-BRM or GFP-BRM-K804R (*Figure 8J*). The fluorescence recovery of mCh-H2B overlapping with GFP-BRM-K804R (7 (± 3) %) was significantly slower than in the presence of GFP-BRM (18 (± 3) %; *Figure 8K*). These results suggest that chromatin remodeling and remodeler release are intrinsically coupled.

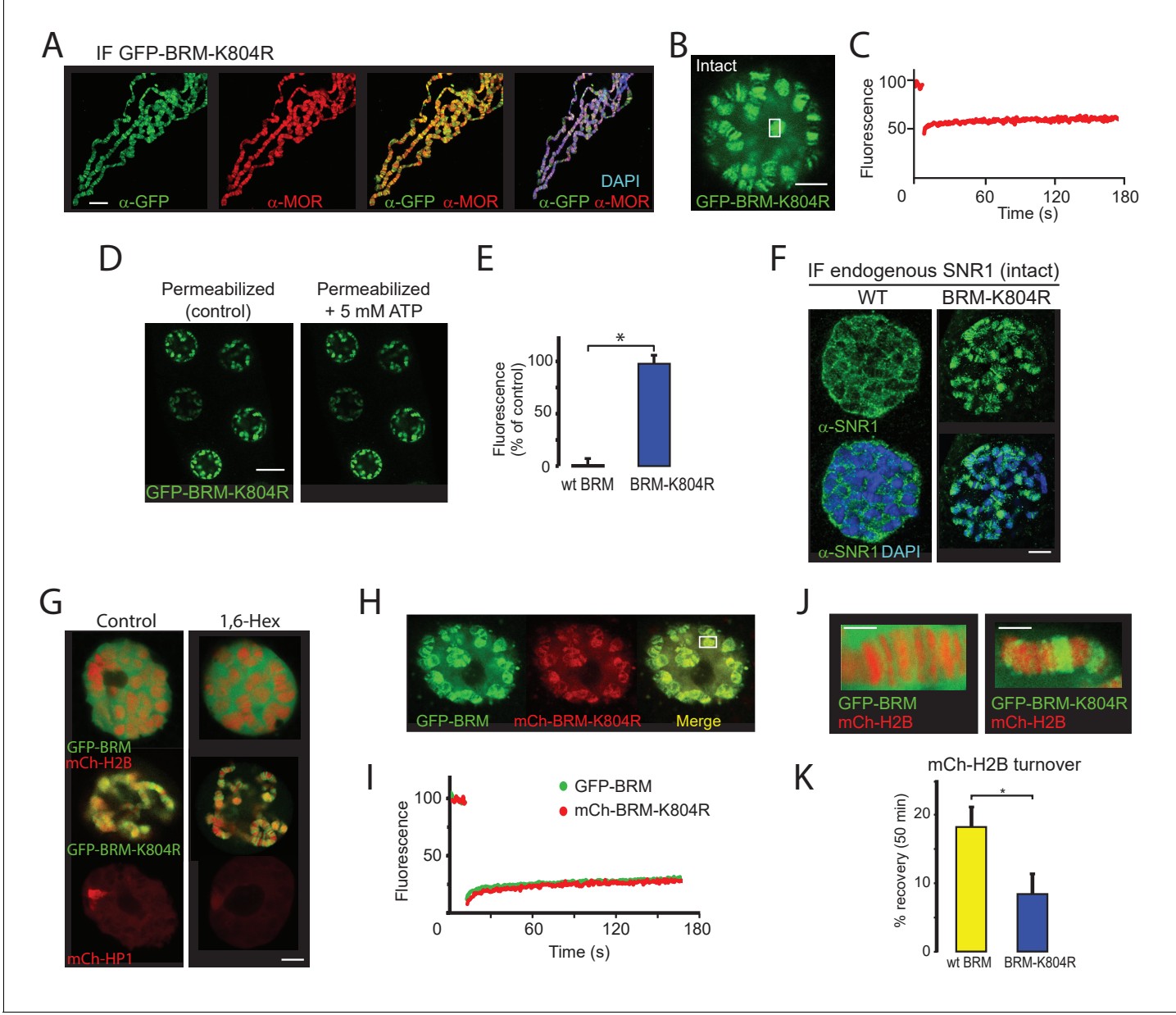

**Figure 8.** BRM-K804R clamps onto polytene chromosomes and impedes histone turnover. (**A**) Distribution of GFP-BRM-K804R on a polytene chromosome spread determined by indirect IF using antibodies against GFP (green) and MOR (red). DNA was visualized by DAPI staining (blue). Scale bar represents 20 μm. (**B**) Confocal image of GFP-BRM-K804R expressing salivary gland. Scale bar represents 5 μm. (**C**) FRAP of GFP-BRM-K804R chromosomal band. Rectangle in panel B indicates the bleached area. (**D**) Confocal images of GFP-BRM-K804R in a permeabilized gland prior or 5 min after the addition of ATP to a final concentration of 5 mM. Scale bar represents 20 μm. (**E**) Comparison of GFP-BRM and GFP-BRM-K804R fluorescence in permeabilized glands 5 min after addition of 5 mM ATP. Fluorescence is expressed as percentage of the intensity prior to ATP addition (mean +/- SD for n = 4 (GFP-BRM)) or 8 (GFP-BRM-K804R). Asterisk indicates a significant difference (p < 0.05). (**F**) GFP-BRM-K804R determines chromatin association of endogenous SNR1. Confocal IF images of whole mount nuclei from formaldehyde fixed salivary glands expressing either wt GFP-BRM or GFP-BRM-K804R using antibodies against SNR1 and DAPI. Scale bar represents 5 μm. (**G**) Confocal images of salivary gland polytene nuclei expressing mCh-H2B, GFP-BRM, GFP-BRM-K804R, or mCh-HP1 in either the absence or presence of 5% 1,6-Hex for 5 min. (**H**) BRM-K804R traps wt BRM onto polytene chromosomes. Confocal image of a nucleus in cultured salivary gland expressing both GFP-BRM and mCh-BRM-K804R. Scale bar represents 5 μm. (**I**) Two color FRAP of GFP-BRM (green) and mCh-BRM-K804R (red) chromosomal band. Rectangle in panel H indicates the bleached area. (**J**) Zoom of polytene chromosome in glands that co-express mCh-H2B and either GFP-BRM or GFP-BRM-K804R. Scale bar represents 2.5 μm. (**K**) BRM-K804R impedes histone turnover. Single chromosomal band FRAP of mCh-H2B co-localized with either GFP-BRM (yellow) or GFP-BRM-K804R (blue). Data are expressed as percentage recovery within 50 min after photobleaching (mean ± S.E.M. for n = 5). Asterisk indicates a significant difference (p < 0.05). Images are representative for polytene nuclei in salivary glands obtained from multiple larvae and independent crosses.

## Discussion

We studied the role of ATP-hydrolysis during exploration of the genome by (P)BAP chromatin remodelers. We took advantage of the parallel amplification of the *Drosophila* genome in salivary gland polytene chromosomes to monitor (P)BAP interactions with endogenous targets in real time. This allowed us to distinguish between nucleoplasm and natural loci in interphase chromosomes by live-cell imaging, which is not possible in diploid cells. We found that a surprisingly small portion of (P)BAP (~5%) is associated with chromatin at any given time. This is caused by the coupling of ATP-hydrolysis during nucleosome remodeling to (P)BAP release. Consequently, (P)BAP acts through a continuous transient probing of the genome (*Figure 9A*). (P)BAP mainly interacts with chromatin in non-condensed interbands or puffs. Conversely, (P)BAP levels within highly condensed polytene bands are much lower than in the nucleoplasm. We note that the loss of (P)BAP does not affect chromatin condensation and the polytene banding pattern (*Figure 1—figure supplement 1A*). Based on these observations, we postulate that chromatin condensation excludes (P)BAP, thereby reducing its genomic search space. At the height of *E74* and *E75* transcription, a substantial portion of RNAPII is retained locally in what Lis and colleagues named a transcription compartment (*Yao et al., 2007*). On the same chromatin puffs, however, PBAP exchanges rapidly (within ~2–4 s) without a substantial immobile fraction. Thus, although essential for expression of *E74* and *E75*, PBAP is not retained within a transcription compartment. The rapid exchange of (P)BAP indicates continuous cycles of nucleosome remodeling. Transient binding to chromatin might explain the relatively poor capture of remodelers by ChIP, as the crosslinking reaction has temporal constraints (*Gelbart et al., 2005*; *Schmiedeberg et al., 2009*). Indeed, a mutation in the ATP-binding pocket increased the residence time on chromatin and concomitantly enhanced ChIP signals of BRM-K804R. Consistent with our earlier developmental genetic analysis (*Chalkley et al., 2008*), we found that PBAP, but not BAP, is required for Ec-regulated gene expression. This finding expands the functional diversification between BAP and PBAP remodeling complexes, which is to a large extent determined by their signature subunits (*Mohrmann et al., 2004*; *Moshkin et al., 2007*; *Chalkley et al., 2008*;

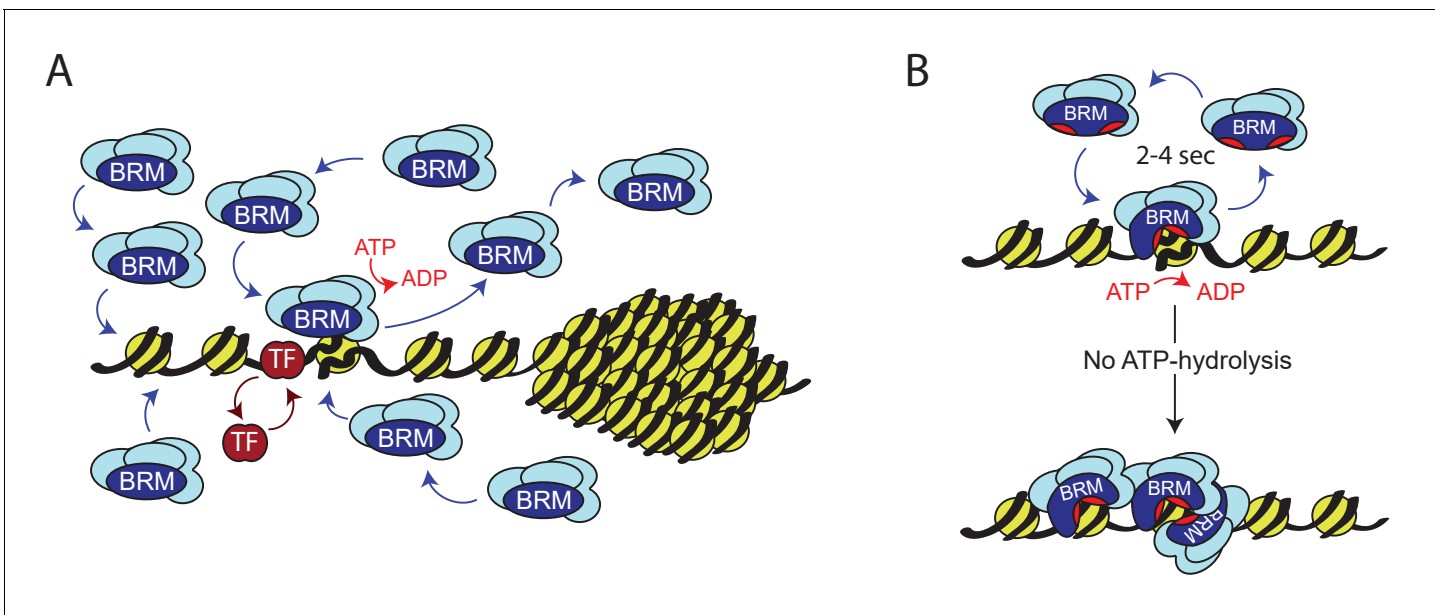

**Figure 9.** Model for ATP-dependent chromatin remodeler dynamics. (A) (P)BAP remodelers act through a continuous transient exploration of the genome that is driven by the coupling between ATP-hydrolysis and remodeler release from chromatin. In contrast, histone exchange is limited and more than 2 orders of magnitude slower. Chromatin condensation leads to the exclusion of (P)BAP, thus reducing its genomic search space. Sequence-specific transcription factor (TF). (B) ATP-hydrolysis stimulates remodeler release, but is not required for binding to chromatin. In solution, the two domains that harbor the ATP-binding motifs (red pockets) are oriented away from one another, precluding ATP-binding. Upon DNA-binding, these core domains rotate towards each other, forming a functional ATP-binding pocket (*Xia et al., 2016*). ATP-hydrolysis not only enables nucleosome remodeling but also increases the probability of remodeler release. This drives rapid cycles of remodeling and remodeler recycling. Loss of ATP-hydrolysis results in retention and clustering of (P)BAP on chromatin, and reduced histone turnover. For details see main text.

*Moshkin et al., 2012*). We detected the fly homolog of GLTSCR1 in GFP-BRM IPs from S2 cells (*Figure 4B*). Indeed, a recent study identified and characterized the fly BRM complex that is orthologous to mammalian GBAF/ncBAF (*Barish et al., 2020*). Thus, although GBAP appears to be mainly expressed in the nervous system, it might contribute to our analysis of BRM dynamics. We note that the dynamic characteristics of SNR1, D4 and BRD7, which are all absent from GBAP, is highly similar to that of BRM.

Histones displayed only limited exchange, which was more than 2 orders of magnitude slower than that of (P)BAP. FRAP analysis of GFP-H2B in polytene chromosomes showed that it takes about 40 min to obtain a ~18% recovery (*Figure 5H*). These results agree with the notion that histone eviction is restricted to small regulatory regions of the genome and is associated with the placement of variant histones (*Brahma and Henikoff, 2020*; *Cakiroglu et al., 2019*; *Deal et al., 2010*; *Pillidge and Bray, 2019*). Indeed, it is difficult to envision how the limited amount of free canonical histones available in the nucleoplasm could support substantial turnover of chromosomal histones. A comparison of DNA accessibility and histone occupancy during transcriptional induction revealed that changes in the former predominate the latter (*Mueller et al., 2017*). Likewise, several recent studies indicated that promoter chromatin harbors so called fragile nucleosomes with accessible DNA, which depend on chromatin remodeler activity (*Brahma and Henikoff, 2020*). These observations suggest that the prevalent outcome of chromatin remodeling in vivo is restructuring or sliding of nucleosomes rather than eviction.

Vital imaging of fluorophore-tagged histones recapitulated the polytene banding pattern and local chromatin condensation. This enabled us to match the GFP-H2B density distribution in living cells with chromatin states derived from histone marks and DNA accessibility (*Eagen et al., 2015*; *Filion et al., 2010*; *Kharchenko et al., 2011*; *Zykova et al., 2018*). Our analysis indicated that DNA in gray bands is on average ~6-fold more condensed than that in interbands, whereas heterochromatic DNA can be up to ~150-fold more condensed. The level of gray band condensation measured by live-cell imaging agrees well with the value derived from HiC analysis of diploid Kc167 cells (*Eagen et al., 2015*), whereas our imaging-based estimate for black chromatin condensation is ~5-fold higher. We suspect that this discrepancy is caused by the very high degree of chromatin condensation at the chromocenter and telomeres of polytene chromosomes. Indeed, measurements of black chromatin bands flanking the *E74* and *E75* loci, suggested a ~25-fold condensation compared to open chromatin, which falls in the range derived from HiC in diploid cells (*Figure 3E*; *Eagen et al., 2015*). Our real time measurements of histone density in polytene chromosomes in living cells complements other methods to estimate condensation that all depend on chromatin fixation (*Eagen et al., 2015*; *Ou et al., 2017*; *Zykova et al., 2018*). Collectively, these studies suggest that euchromatin and heterochromatin are formed by the same basic nucleosomal array, but at different degrees of condensation.

Although central to the function of Snf2 remodelers, our understanding of the role of ATP hydrolysis in the nucleosome remodeling cycle remains incomplete. A mutation in the ATP-binding pocket of BRM caused up to 5-fold stronger chromatin binding and a ~2-times slower turnover of BRM-K804R in S2 cells. In polytene cells, BRM-K804R clamped down onto chromatin and displayed no measurable turnover. Likewise, in permeabilized salivary gland cells that lack ATP, wt BRM accumulated onto polytene chromosome interbands. Chromatin binding persisted for over 1 hr, but following the addition of ATP, BRM released immediately. Most likely, the high local density of target loci in polytene chromosomes amplifies the effect of prolonged chromatin binding by BRM. We speculate that in the absence of ATP hydrolysis, slow release combined with rapid re-binding to a paired chromatid, might results in the local trapping of (P)BAP on polytene chromosomes. We note that under normal physiological conditions, ATP concentrations in cells would not become depleted to a level (< 1 mM) that would impair remodeler function. Finally, our results argue against a role for liquid-liquid phase separation in the clustering of (P)BAP in the absence of ATP-hydrolysis. Local (P)BAP accumulation is insensitive to the disruption of weak hydrophobic interactions by 1,6-hexanediol. Moreover, FRAP analysis showed that clustered BRM has a strongly reduced mobility, indicating it does not form a condensate. We consider prolonged chromatin-binding the most plausible cause of local accumulation of BRM-K804R, or wt BRM in the absence of ATP.

Studies in diploid cells often equate the formation of low-mobility aggregates of a fluorophore-tagged transcription factor with persistent chromatin binding and functional activity. However, given that chromatin typically only occupies about 3% of the nuclear volume, this interpretation remains

speculative. E.g., in a study on human ISWI remodelers, an increase in the immobile fraction upon DNA damage was rationalized as an increase in activity on chromatin (*Erdel et al., 2010*). Moreover, it was concluded that loss of ATP-binding reduced chromatin association and increased mobility. In contrast, our results with BRM showed that chromatin binding is independent of ATP, but that its release is stimulated by ATP-hydrolysis. At the peak of PBAP- and EcR-dependent transcription of *E74* and *E75* genes, both retained a rapid exchange, indicating ceaseless cycles of nucleosome remodeling. Consistent with this notion, the use of small-molecule inhibitors recently showed that chromatin accessibility required continuous remodeler activity (*Iurlaro et al., 2021*). Transient remodeler dynamics enables competition at regulatory elements between epigenetic regulators with opposing effects on chromatin structure (*Bracken et al., 2019*; *Cenik and Shilatifard, 2021*; *Kubik et al., 2019*; *Mohd-Sarip et al., 2017*).

There are both parallels and differences in the roles of ATP in either chromatin remodeling or in RNA-duplex unwinding by DEAD-box helicases. ATP-hydrolysis is dispensable for RNA-binding and duplex unwinding by DEAD-box helicases, but is required for fast enzyme release from RNA (*Liu et al., 2014*). However, the formation of long-lived RNA-DEAD-box helicase complexes requires ATP-binding but not its hydrolysis. In contrast, chromatin binding by BRM is independent of ATP. The ATP hydrolysis-fueled transient probing of the genome by BRM we described here, dovetails well with structural observations on Swi2/Snf2 remodelers (*Xia et al., 2016*). When Snf2 is free in solution, the two domains that harbor the ATP-binding motifs are oriented away from one another and Snf2 cannot bind ATP. Binding to naked DNA or nucleosomal DNA induces a reorientation of the two ATPase lobes so that they now form a functional ATP-binding pocket. Combined with our results, these structural observations suggest a model in which nucleosome binding occurs independent of ATP but induces the formation of an ATP-binding pocket (*Figure 9B*). ATP hydrolysis can now fuel nucleosome remodeling coupled to an increased probability of remodeler release, resulting in rapid cycles of nucleosome remodeling and remodeler recycling. The absence of ATP-hydrolysis leads to prolonged chromatin binding of the remodeler and impedes nucleosome remodeling.

Our observation that ATP hydrolysis promotes remodeler release from chromatin raises the question of how many translocation steps a single remodeler takes before dissociation from a nucleosome in vivo. Previous single molecule studies on the RSC remodeler determined that an ATP concentration above 1 mM is required for optimal translocation speed and processivity (*Zhang et al., 2006*; *Sirinakis et al., 2011*). This value derived from in vitro experiments is remarkably similar to our in vivo estimates. Note that many biophysical studies on remodelers use much lower ATP concentrations to slow down the reaction speed and determine individual kinetic steps (see e.g. *Harada et al., 2016*). In the presence of 1 mM ATP in vitro, RSC translocated at a speed of 25 bp/s on naked DNA and ~13 bp/s on nucleosomes (*Zhang et al., 2006*; *Sirinakis et al., 2011*). Enzyme processivity in these experiments was ~35 bp on naked DNA and (albeit it with a widespread) on average ~100 bp per round of translocation on nucleosomes. Stopped-flow pre-steady state kinetic analysis of RSC translocation along naked DNA suggested a series of uniform steps yielding an average translocation distance of 20 bp before dissociation (*Fischer et al., 2007*). The (P)BAP turnover time of ~2–4 s in our in vivo FRAP experiments agrees well with the in vitro single molecule analyses and is compatible with multiple kinetic steps and substantial nucleosome remodeling prior to remodeler disengagement. Remodeler release might be considered as a probability during each ATPase cycle. Full remodeling, sliding or even eviction may require the cooperation of a cascade of remodelers acting subsequently.

BRM-K804R clamps to chromatin and impedes histone turnover, emphasizing that remodeling and remodeler recycling are closely integrated. Surprisingly, BRM-K804R also traps wt BRM onto polytene chromosomes. Thus, reflecting early genetic studies (*Elfring et al., 1998*), BRM-K804R behaves as a molecular dominant mutant. The retention of wt BRM onto chromatin by BRM-K804R suggests that BRM complexes might interact on chromatin. Consequently, remodeling might involve a nucleosome hand-off mechanism between different BRM complexes. Additionally, translocation by one BRM complex might help to push another one off the chromatin. This way, the persistent chromatin binding of BRM-K804R might create a 'traffic jam' that blocks the release of wt BRM.

In summary, our visualization of BRM remodelers engaging their endogenous genomic targets revealed highly transient and dynamic interactions. (P)BAP binds chromatin independent of ATP. However, ATP-hydrolysis couples nucleosome remodeling to (P)BAP release, resulting in rapid cycles of remodeling and remodeler turnover. Remodelers might interact on chromatin, as suggested by

the trapping of wt BRM by the ATP-binding mutant BRM-K804R. Importantly, an accompanying report by Wu and colleagues, studying budding yeast remodelers through live-cell single molecule tracking, also revealed the key role of the ATPase in promoting dissociation from chromatin and fast remodeler kinetics (*Kim et al., 2021*). Collectively, these in vivo results provide a framework for understanding remodeler function in genome regulation.

# Materials and methods

## Cloning procedures

Full-length cDNAs encoding BRM, SNR1, D4, BRD7, EcR, RBP2, RBP3, H2A, and H2B were cloned into pENTR/TEV/D-TOPO by TOPO cloning according to the manufacturer's protocol, resulting in a series of pENTR vectors, which was used to generate derivative constructs. BRM-K804R was constructed via PCR-based site-directed mutagenesis of pENTR-Brm using standard procedures. To generate vectors that express eGFP- or mCherry (mCh)- fusion proteins, the appropriate coding sequences were recombined into the destination vectors pTGW or pTChW using LR ClonaseII (Invitrogen) according to the manufacturer's protocol. pTGW was obtained from the *Drosophila* Genomics Resource Center (DGRC). The pTChW vector was constructed from pTGW by replacing the GFP coding sequence with mCh using the NcoI and AgeI restriction sites. The constructs for expression of GFP-BRM or GFP-BRM-K804R under the control of the metallothionein (Mt) promoter in S2 cells were generated by LR Clonase II-mediated recombination of BRM and BRM-K804R into pMGW. pMGW is a modified version of pAGW (DGRC) in which the actin promoter has been replaced by the Mt promoter. The Mt promoter was PCR-amplified from pMT/V5-HisA (Invitrogen) and ligated into pAGW using BglII and EcoRV restriction sites. The integrity of all constructs we generated was verified by DNA sequencing of the entire coding sequence and flanking regions.

## *Drosophila* stocks

To generate transgenic fly lines, the appropriate constructs were injected in *Drosophila yw* embryos or $w^{1118}$ and balanced transformants were isolated by BestGene Inc (Chino Hills, USA) and further analyzed in our lab. GFP-PC flies were a gift from Renato Paro (*Dietzel et al., 1999*) and the mCh-HP1 line was a gift from Jean-Michael Gilbert and Francois Karch (University of Geneva). UAS-NLS-NES$^{P12}$-GFP (stock number 70330) and Sgs3-GAL4 (salivary gland driver, stock number 6870) fly lines were obtained from the Bloomington *Drosophila* Stock Center (https://bdsc.indiana.edu). The *polybromo33.2* and *bap170$^{kim1}$* mutants have been described (*Chalkley et al., 2008*). All stocks were maintained on standard corn medium at 18℃. All crosses and experiments were carried out at room temperature. A list of *Drosophila* lines is provided in *Supplementary file 1*.

## Generation of GFP-BRM expressing cell lines

Schneider 2 cells (obtainded from ATCC; ATCC Cat# CRL-1963, RRID:CVCL_Z232) were maintained in Schneiders *Drosophila* medium containing 10% fetal calf serum at 25℃. S2 cells were transfected with cellfectin II reagent (Thermo Fisher Scientific) according to the manufacturer's protocol. Cells tested negative for mycoplasma contamination. To enable drug selection, pCoBlast (Invitrogen) was cotransfected with either pMGW-Brm or pMGW-Brm-K804R in a ratio of 1:20. After 24 hr, transfected cells were selected with 25 µg/ml blasticidin. Following the establishment of polyclonal stable lines, cells were cultured in the presence of 5 µg/ml blasticidin.

## ChIP-qPCR and RT-qPCR procedures

Schneider 2 (S2) cells were maintained in Schneider's *Drosophila* medium containing 10% fetal calf serum at 25℃. GFP-BRM and GFP-BRM-K804R expression in S2 cells was induced for three days by addition of CuSO4 (500 µM final concentration). Cells were fixed with 1% formaldehyde for 10 min. Fixation was stopped with 100 mM glycine. Next, cells were washed twice with PBS and lysed in 0.75% SDS, 10 mM EDTA, 50 mM Tris pH 8.0, supplemented with protease inhibitors (1 µg/ml pepstatin A, 1 µg/ml leupeptin, 1 µg/ml aprotinin, 0.2 mM AEBSF), which were present in all subsequent buffers until elution. The volume of the lysates was measured and SDS concentration was adjusted to 0.6%. Cross-linked chromatin was sheared to 200–300 bp of DNA length in a Bioruptor UCD-200 sonicator. Chromatin concentration was measured in a NanoDrop spectrophotometer (Thermo

Fisher Scientific) and equal amounts were used as input for each ChIP. Chromatin was diluted 10-fold with 1% Triton X-100, 2 mM EDTA pH 8.0, 150 mM NaCl and 20 mM Tris pH 8.0, cleared by centrifugation and incubated overnight at 4˚C with either 1.5 µl anti-GFP (Abcam #290) or mock anti-bodies. Next morning, 10 µl preblocked Protein A Sepharose (GE Healthcare) was added and incubated for 2 hr, followed by four washes with wash buffer (20 mM Tris-HCl pH8.0, 2 mM EDTA pH 8.0, 0.1% SDS, 1%Triton X-100) containing 150 mM NaCl and 1x with wash buffer containing 500 mM NaCl. For DNA elution, samples were incubated in 0.1 M NaHCO$_3$, 1% SDS, and 0.5 mg/ml proteinase K for 2 hr at 37˚C and overnight at 65˚ C. DNA was subsequently purified by phenol/chloroform extraction, followed by ethanol precipitation. Immunopurified chromatin samples were analyzed by real-time PCR, using the comparative Ct method (*Mutskov and Felsenfeld, 2004*). The results of three independent biological replicates were averaged and the standard deviation was determined. For prepupal analysis, *polybromo33.2* and *bap170^{kim1}* mutants mutant lines were rebalanced with GFP-marked balancer chromosomes. GFP-negative mutant prepupae and wt prepupae were collected at 2 hr intervals from the moment of pupariation (*t* = 0) for 12 hr. RNA was extracted using TRIzol. Knockdown of (P)BAP subunits in S2 cells was performed as described (*Chalkley et al., 2008*; *Moshkin et al., 2007*; *Moshkin et al., 2012*). Twenty-four hr after treatment with dsRNA, 20-hydroxyecdysone was added to a final concentration of 20 µM. Cells were harvested 24 hr later and RNA was extracted using TRIzol. Knockdowns were performed biological triplicates. RNA levels in prepupae or S2 cells were analyzed by first-strand cDNA synthesis with Superscript II reverse transcriptase (Invitrogen) and subsequent quantitative PCR (qPCR) with SYBR green I using a MyiQ single-color real-time PCR detection system (Bio-Rad). Analysis of the reverse transcription (RT)-qPCR data was performed using the $2^{-\Delta\Delta CT}$ method (*Livak and Schmittgen, 2001*). CG11874 was used as an internal control mRNA. A list of primers used is provided in *Supplementary file 2*.

## Live-cell imaging and FRAP experiments

GFP-BRM and GFP-BRM-K804R expressing S2 cells were grown on glass coverslips and experiments were performed in Schneider *Drosophila* medium containing 10% FCS. Intact salivary glands expressing eGFP or mCh-tagged proteins were excised in PBS, rinsed and transferred to an incubation chamber containing Grace's insect medium diluted 5: one with water (*Yao et al., 2008*). Confocal images (1024 x 1024 pixels) of the nuclei were acquired at room temperature using a Leica TCS SP5 confocal microscope (Argon 488 nm and HeNe 594 nm laser lines), an 63x HCX PL APO CS 1.4 NA oil-immersion objective and Leica LAS-AF acquisition software. Images were processed and analysed using Fiji/ImageJ software (https://imagej.net). FRAP experiments were performed on a Leica TCS SP5 confocal microscope using Leica LAS-AF data acquisition software and a 40x HCX PL APO CS 1.3 NA (salivary glands) or a 63x HCX PL APO CS 1.4 NA (Schneider 2 cells) oil-immersion objective. For Schneider 2 cells, a strip of 128 x 16 pixels across the nucleus was bleached by 100% laser power of the Ar 488 nm laser line. Pre- and post-bleaching images (128 x 128 pixels, total of 20 and 200 images, respectively) were collected with 70 ms intervals and the average pixel intensity of both the bleached area and the whole nucleus was determined. Fluorescence recovery curves were full-scale normalized, double exponential fitted and the half-time of fluorescence recovery (t$_{1/2}$) and mobile fraction were determined using the EasyFRAP online FRAP analysis tool (https://easyfrap.vmnet.upatras.gr; *Koulouras et al., 2018*). Half-nucleus FLIP-FRAP and (dual color) single-band FRAP experiments on salivary glands nuclei were performed in Grace's insect medium diluted 5:1 with water at room temperature. Images (512 x 512 pixels) were acquired using the Ar 488 nm (eEGF) and/or HeNe 594 nm (mCh) laser lines with an interval of 648 ms. Photobleaching was obtained by 100% laser power of the Ar 488 nm (eGFP) and/or 100% laser power of the DPSS 561 nm (mCh). For half nucleus FLIP-FRAP, the average pixel intensity of both the unbleached and bleached areas were determined, corrected for background fluorescence and expressed as percentage of the average pre-bleaching value. For single band FRAP, the average pixel intensity of the bleached area and of the whole nucleus were determined for a series of 10 pre-bleach and 250 post-bleach images and corrected for background fluorescence. Recovery curves were double exponential fitted and the half-time of fluorescence recovery (t$_{1/2}$) and mobile fraction were determined using EasyFRAP online. The intensity of GFP-H2B was measured across 10 nuclei, nine confocal slices per nucleus. Brightness was auto-adjusted using Fuji/imageJ software (https://imagej.net). GFP-H2B pixel intensities were binned into four classes, corresponding to background, white, gray, and black chromatin, using the following tresholds: background 0–82; Interband/white chromatin, 83–148;

(Low-intensity bands/gray chromatin 149–224) and (high-intensity/black chromatin >224). For raw data see *Figure 1—source data 1*.

## Permeabilization with digitonin

Freshly isolated salivary glands were transferred to a glass multiwall plate and permeabilized essentially as described previously (*Myohara and Okada, 1987*). Permeabilization was started by replacing the medium with MTB1 buffer (15 mM $KH_2PO_4$; 50 mM KCl, 15 mM NaCl, 7.5 mM $MgCl_2$, 1% PEG 6000, pH = 7.0, KOH) containing 0.01% digitonin. After 20 min of permeabilization and three subsequent washes with MTB1 buffer, the glands were transferred to a glass Sigma cloning ring containing MTB1 buffer mounted on the coverslip of the incubation chamber. Confocal imaging was performed in MTB1 medium as described above. For immunocytochemistry, the permeabilized glands were washed with MTB1 buffer (three times), incubated for 5 min in MTB1 either in the absence or presence of ATP (10 mM), and subsequently fixed and processed for immunocytochemistry as described.

## Immunofluorescence microscopy

Immunolocalization of proteins on *Drosophila* salivary gland polytene chromosome spreads was performed essentially as described previously (*Mohrmann et al., 2004*), using the indicated primary antibodies. Slides were mounted in mounting medium containing 4',6'-diamidino-2-phenylindole (DAPI) counter stain (Vector Laboratories). For preparations of polytene chromosomes in the presence of ecdysone hormone, dissected salivary glands were incubated for 1 hr in Grace's medium (11605–045 Invitrogen), diluted 5:1 with $H_2O$, containing 20 µM 20-hydroxyecdysone. Endogenous proteins in intact, dissected salivary glands were immunolocalized by incubating the glands in MTB1 buffer (15 mM $K_2HPO_4$, 50 mM KCl, 15 mM NaCl, 7.5 mM MgCl2, 1% PEG6000, pH7.0 with KOH) in the presence or absence of digitonin 0.01% or 10 mM ATP. Glands were then washed three times with MTB1 buffer before fixation with 3.7% formaldehyde in TBST (50 mM Tris-HCl pH8, 150 mM NaCl, 0.1% Triton X-100) at room temperature for 10 min. Preparations were blocked with TBST containing 5% FCS. Next, they were incubated with primary antibodies, diluted in TBST containing 1% BSA, overnight at 4°C. Following multiple washes with TBST buffer, they were incubated with secondary Alexa Fluor antibodies, washed again and finally mounted in Vectorshield mounting medium (Vector Laboratories) containing DAPI. A list of antibodies used is provided in *Supplementary file 3*.

## Biochemical procedures

Immunoprecipitations and immunoblotting experiments were performed using standard methods (*Chalkley and Verrijzer, 2004*; *Chalkley et al., 2008*; *Harlow and Lane, 1998*). All procedures were on ice or at 4°C. S2 cells expressing either GFP-BRM or GFP-BRM-K804R were harvested 3 days after the addition of CuSO4 to a final concentration of 500 µM. Following a freeze-thaw step, whole cell extracts were prepared in HEMG/150: 25 mM HEPES-KOH pH7.6, 0.1 mM EDTA, 12.5 mM MgCl2, 10% glycerol, containing 150 mM KCl, 0.5% NP40 and a cocktail of protease inhibitors: 1 µM pepstatin, 1 µM aprotinin, 1 µM leupeptin, 0.2 mM 4-(2-Aminoethyl) benzenesulfonyl fluoride hydrochloride (AEBSF). After rotation for 20–30 min at 4°C, the extracts were sheared through an insulin syringe. Extracts were then diluted with HEMG/150 to obtain a final NP40 concentration of 0.1%, and then clarified by centrifugation. For each immunoprecipitation, 20–25 µl of Chromotek GFP-trapA beads were used for 1–4 mg total protein. Extracts were incubated with beads for 1 hr at 4°C. The beads were then washed extensively with the following buffers: HEMG/100/0.1% NP40; HEMG/600/0.1% NP40; HEMG/100/0.01% NP40 and finally with HEMG/100 without NP40. Co-immunoprecipitation reactions of proteins from *Drosophila* salivary glands were carried out as follows: *Drosophila* salivary glands were dissected and collected in PBS. Following collection, PBS was removed and replaced with urea extraction buffer (HEMG/150 mM KCl, containing 0.1% NP40, 1M urea and 1 mM DTT) at a ratio of ~1 µl urea extraction buffer per pair of glands. Collected glands were snap frozen in liquid nitrogen. Once sufficient glands were collected, samples were pooled and next crushed with an eppie-pestle, followed by shearing by passage through an insulin needle. Extracts were then clarified by centrifugation. For immunoprecipitation assays, the equivalent of ~300 dissected salivary glands was incubated with 100 µg of anti-MOR antibodies cross-linked to Protein A-Sepharose beads (or control beads coated with pre-immune serum) for 2–3 hr at 4°C. Beads were washed sequentially

with 2x HEMG/500/0.1% NP40 and 1x HEMG/100/0.01% NP40; all buffers contained protease inhibitors. Bound proteins were eluted by boiling for 1 min in SDS-loading buffer. Proteins were resolved by SDS-PAGE followed by immunoblotting. Each lane on the IP-Western analysis represents the equivalent of ~30 animals.

## Mass spectrometric analysis

For mass spectrometric analysis, proteins were on-bead subjected to reduction with dithiothreitol, alkylation with iodoacetamide and digested with trypsin (sequencing grade; Promega). Nanoflow liquid chromatography tandem mass spectrometry (nLC-MS/MS) was performed on an EASY-nLC coupled to an Orbitrap Fusion Tribid mass spectrometer (Thermo), operating in positive mode. Peptides were separated on a ReproSil-C18 reversed-phase column (Dr Maisch; 15 cm × 50 μm) using a linear gradient of 0–80% acetonitrile (in 0.1% formic acid) during 90 min at a rate of 200 nl/min. The elution was directly sprayed into the electrospray ionization (ESI) source of the mass spectrometer. Spectra were acquired in continuum mode; fragmentation of the peptides was performed in data-dependent mode by HCD. Raw mass spectrometry data were analyzed with the MaxQuant software suite (*Cox et al., 2009*; version 1.6.7.0) with the additional options 'LFQ' and 'iBAQ' selected. A false discovery rate of 0.01 for proteins and peptides and a minimum peptide length of 7 amino acids were set. The Andromeda search engine was used to search the MS/MS spectra against the Uniprot database (taxonomy: *Drosophila melanogaster*, release January 2019) concatenated with the reversed versions of all sequences. A maximum of two missed cleavages was allowed. The peptide tolerance was set to 10 ppm and the fragment ion tolerance was set to 0.6 Da for HCD spectra. The enzyme specificity was set to trypsin and cysteine carbamidomethylation was set as a fixed modification, while STY phosphorylation was set as a variable modification. Both the PSM and protein FDR were set to 0.01. In case the identified peptides of two proteins were the same or the identified peptides of one protein included all peptides of another protein, these proteins were combined by MaxQuant and reported as one protein group. Before further statistical analysis, known contaminants and reverse hits were removed. To determine the relative abundances of proteins within a sample, the iBAQ intensities were compared.

## Quantification and statistical analysis

Statistical parameters including the value of n, % recovery, $t_{1/2}$ and mobile fraction (mean +/- SD) are indicated in the figures or figure legends. Data is judged to be statistically significant when $p < 0.05$ by two-tailed Student's t test and is indicated by an asterisk in the figures.

# Acknowledgements

We thank Jean-Michel Gilbert, Francois Karch, and Renato Paro for the gift of fly stocks and the Erasmus Optical Imaging Centre (Gert van Cappellen, Adriaan Houtsmuller and Gert-Jan Kremers) for technical assistance and helpful discussions. We also acknowledge Olaf Voets, Adrie Verhoeven and Prasanth Kumar for their assistance in generating various constructs during the initial stages of this project. We thank Jesper Svejstrup and Aniek van der Vaart for valuable comments on the manuscript. Finally, we thank Carl Wu and Jee Min Kim for sharing their results prior to submission for publication. This work was supported in part by a network grant from FOM ('DNA in action: Physics of the genome').

# Additional information

## Funding

| Funder | Grant reference number | Author |
| --- | --- | --- |
| Netherlands Organisation for Scientific Research | DNA at Work | C Peter Verrijzer |

The funders had no role in study design, data collection and interpretation, or the decision to submit the work for publication.

## Author contributions
Ben C Tilly, Conceptualization, Data curation, Software, Formal analysis, Validation, Investigation, Visualization, Methodology, Writing - original draft, Writing - review and editing; Gillian E Chalkley, Data curation, Validation, Investigation, Visualization, Methodology, Writing - review and editing; Jan A van der Knaap, Conceptualization, Data curation, Validation, Investigation, Methodology, Writing - review and editing; Yuri M Moshkin, Data curation, Formal analysis, Supervision, Validation, Investigation, Visualization, Methodology; Tsung Wai Kan, Investigation, Methodology; Dick HW Dekkers, Data curation, Formal analysis, Investigation; Jeroen AA Demmers, Data curation, Formal analysis, Supervision, Investigation, Methodology, Writing - review and editing; C Peter Verrijzer, Conceptualization, Resources, Data curation, Formal analysis, Supervision, Funding acquisition, Investigation, Methodology, Writing - original draft, Project administration, Writing - review and editing

## Author ORCIDs
C Peter Verrijzer https://orcid.org/0000-0002-6476-3264

## Decision letter and Author response
Decision letter https://doi.org/10.7554/eLife.69424.sa1
Author response https://doi.org/10.7554/eLife.69424.sa2

## Additional files

### Supplementary files
- Supplementary file 1. List of *Drosophila* lines.
- Supplementary file 2. List of Primers.
- Supplementary file 3. List of Antibodies.
- Transparent reporting form

### Data availability
The mass spectrometry proteomics data have been deposited to the ProteomeXchange Consortium (http://www.proteomexchange.org/) via the PRIDE partner repository with the dataset identifier PXD025474. All data generated or analyzed during this study are included in the manuscript and supporting files. Source data files have been provided for Figures 1, 2 and Figure 2—figure supplement 1.

The following dataset was generated:

| Author(s) | Year | Dataset title | Dataset URL | Database and Identifier |
|---|---|---|---|---|
| Verrijzer CP | 2021 | In vivo analysis reveals that ATP-hydrolysis couples remodeling to SWI/SNF release from chromatin | http://proteomecentral. proteomexchange.org/ cgi/GetDataset?ID= PXD025474 | ProteomeXchange, PXD025474 |

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
