## [Decision Letter]

**Acceptance summary:**

ATP-dependent chromatin remodelers scan the genome and can open up chromatin (to enable access of transcription or DNA repair factors) via nucleosome movement or eviction, and then release from that location to conduct new work. Currently, it is challenging to quantify remodeler-nucleosome dynamics in vivo and the role of ATP binding and hydrolysis in the retention and recycling of remodelers on chromatin/nucleosomes. Here, Tilly and co-workers address these questions using elegant live-cell imaging, applied to monitor the SWI/SNF-family Brahma (BRM) remodeler in *Drosophila* polytene nuclei, and show that fast kinetics and likely cooperation between remodelers enables chromatin opening. This manuscript will be of considerable interest to the remodeler and chromatin communities.

**Decision letter after peer review:**

Thank you for submitting your article "in vivo Analysis Reveals that ATP-hydrolysis Couples Remodeling to SWI/SNF Release from Chromatin" for consideration by *eLife*. Your article has been reviewed by 3 peer reviewers, and the evaluation has been overseen by Sebastian Deindl as the Reviewing Editor and Kevin Struhl as the Senior Editor. The following individual involved in review of your submission has agreed to reveal their identity: Bradley R Cairns (Reviewer #2).

Essential revisions:

1) The study presents a wealth of data in a rather compact format. As a result, some arguments appear a bit abbreviated, some methods could be explained more thoroughly, and some controls are lacking for a more stringent argumentation. By addressing the corresponding reviewers' comments below, this highly interesting manuscript can be significantly improved.

Although additional data could further strengthen the manuscript, we feel that the points raised by the reviewers can be adequately addressed with textual revision and by consolidating and more rigorously quantifying the data, and more explicitly stating controls, etc. Note that no new experimentation is called for.

2) One worry is that the tagged proteins are not functional, but there may be good arguments as to why this is less of an issue (e.g., the effect of ATP depletion and KR mutation). The authors should make these arguments in the paper.

3) All three reviewers felt that the clarity of the presentation could be improved by bringing together the sections related to ATP binding and hydrolysis. We strongly recommend such a change in the order of presentation.

*Reviewer #1:*

Strength: Chromatin remodelers play critical roles in mobilization of nucleosomes to modulate chromatin-templated functions, but the molecular mechanisms underlying the dynamics of remodelers interacting with chromatin remains incompletely understood. By using cellular imaging, this study reports that chromatin remodeler mutant, which is defective in ATP-binding, reside longer at chromatin than wild-type one, and further indicates that ATP-hydrolysis stimulates remodeler release from chromatin. By leveraging the characteristic of polytene chromatin being visible under fluorescence microscopy, the authors study that the dynamics of chromatin remodelers at native loci, providing further evidence that chromatin remodelers dynamically interact with chromatin with turn-over time less than 5 sec. Overall, these studies indicate that chromatin remodelers rapidly probe the genome and ATP-binding and hydrolysis play important roles in release of remodelers from chromatin.

Weakness: There are a few studies that have reported that remodelers dynamically exchange with chromatin. The novelty of this study is the discovery of the roles of ATP-binding and hydrolysis on the dissociation of remodelers from chromatin; however, to this reviewer, there is lack of functional significance of the destabilization or mechanistic insights into the destabilization.

1) The authors study the dynamics of BRM-K804R within polytene nuclei (Figure 7). The authors also investigate the dynamics of wild-type BRM at the E74 and E75 loci (Figure 5). It appears to this reviewer that one nice experiment would be to investigate the exchange dynamics of BRM-K804R at these two loci and to study the effects of BRM-K804R on their transcriptional levels. Thus, it would be possible to correlate the binding dynamics and transcription and to provide potential functional significances of rapid exchange of remodelers at chromatin.

2) To this review, it is not easily to follow the logic of the manuscript. It would be helpful to readers if the authors give sufficient reasonings for some experiments. For instance, why do the authors study PBAP and BAP for Ec-induced gene expression (Page 12). To me, the authors should also reorganize some figures and change the order of the presentation.

3) I would suggest that the authors quantify FRAP data to extract kinetic parameters based kinetic modelling. In the current version, when comparing t_1/2_ between different cellular systems, it is not informative.

4) 1,6-hexanediol has been widely used in phase-separation studies. It primarily disrupts hydrophobic interactions. Given that LLPS involves multiple non-covalent interactions, the authors should re-phase some of their conclusions.

5) A schematic that describes PBAP and BAP would help non-remodeler readers in Figure 1.

6) Figure 1e shows difference in FRAP curves between wild-type BRM and BRM-K804R. BRM-K804R forms clusters while BRM not. It would be helpful if FRAP of BRM-K804R is also done in cluster and no-cluster regions.

7) Page 6, the last conclusion sentence seems over-stated since BRM-K804R impacts ATP-binding rather than hydrolysis.

8) Page 10, the second to last sentence, there is no data to support the author's claim.

9) Page 12, the conclusion remark of the first paragraph, there is no data to support hit-and-run' model.

10) To this reviewer, it appears that the retention of BRM-K804R at chromatin at polytene chromatin cannot be simply explained by a slow dissociation rate and a fast on-rate. This is because BRM-K804R can trap wild-type BRM at polytene chromatin and BRM-K804R can be recovery rapidly in S2 cells. Alternative interpretation should be given.

*Reviewer #2:*

ATP-dependent chromatin remodelers must scan the genome for locations to open chromatin (to help provide access to transcription and DNA repair factors) via nucleosome movement or eviction, and then release from that location to conduct new work. Currently, it is challenging to quantify remodeler-nucleosome dynamics in vivo, and the role of ATP binding and hydrolysis on the retention and recycling of remodelers at nucleosomes – aside from the known role of ATP in the process of DNA translocation. These questions are of general interest as chromatin remodelers are intimately involved in the regulation of DNA accessibility for processes such as transcription and DNA repair, and as remodelers must both associate with regions productively yet release properly to search for and conduct the scale of work needed.

Here, Tilly et al., (Verrijzer Lab) and colleagues address these questions using elegant live-cell imaging, applied to monitor the SWI/SNF-family Brahma (BRM) remodeler in *Drosophila* polytene nuclei. This method combined with cleverly chosen reagents, mutants, or physiological contexts, allowed the authors to measure chromatin condensation along with the dynamics governing the interaction of BRM with chromatin. Several conclusions of interest could be reached through the work:

– With the vast majority of BRM is unbound at any given time, BRM remodeler interacts very transiently with chromatin, browsing nucleosomes largely located in decondensed regions.

– ATP binding reduces BRM retention at chromatin, as revealed by characterizing the in vivo dynamics of a Walker A motif mutant.

– ATP hydrolysis increases BRM release from chromatin, as revealed by characterizing the in vivo dynamics in the presence of ADP, or a slow hydrolysable ATP analogue, or a non-hydrolysable ATP ground state mimic.

– When applied to the specific physiological context of highly transcribed ecdysone-induced loci, only the (BRM-containing) PBAP remodeler appears essential; and, remarkably, PBAP fulfills this function without a local retention as shown for RNAPII, but by a rapid exchange.

– Consistent with – and expanding on – the previous conclusions, the presence of the ATP-binding deficient BRM mutant leads to the clustered binding of not only mutated BRM but also wild-type BRM, without the involvement of a phase separation (as 1,6-hexanediol did not affect the punctate/clustered pattern) but likely by accumulation due to excessive retention. Notably, as the authors rightly comment, these interesting results lead to the possibility of a direct cooperation or a hand-off mechanism between different tandem SWI/SNF remodelers, and will deserve further future investigation.

The conclusions listed above are relatively convincingly as they are well supported by the data. The discussion is clearly and thoughtfully written.

Altogether, this work rigorously addresses how ATP binding and hydrolysis cycle of the Brm remodeler, affects the in vivo dynamics of nucleosome browsing, binding and recycling. I think this will be of considerable interest to the remodeler and chromatin community, and recommend publication after revision in *eLife*.

1. To improve the manuscript, I will suggest that the authors consider slightly reorganizing the manuscript in order to bring in close proximity all the sections related to ATP binding and to ATP hydrolysis, which are currently separated by the characterization of the BRM dynamics at the ecdysone-induced loci.

2. I will also ask that the authors note that the reorientation of the two ATPase lobes is not just induced by nucleosomes, but also efficiently elicited by naked DNA, so nucleosomes do not gate that conformational change.

3. I appreciate how the kinetics of the in vivo work was linked to former in vitro studies in the Discussion – and here the authors might consider (in addition to the works cited) how prior pre-steady state stopped-flow kinetic analysis of SWI/SNF-family remodelers and resulting kinetic and probabilistic models (Fischer et al., Biochemistry 2007) also generally align with their work – and how thinking about release as a probability during each ATPase cycle might be of interest.

4. Finally – the identification of a GLTSCR1 ortholog (Figure 1) is really interesting. he authors mention this but do not explore more deeply. This affects the work in two ways – the possible existence of a GBAF/ncBAF related complex in flies, and that this complex may be participating in the processes observed. I understand that a full characterization of a possible separate fly GBAF/ncBAF complex (in addition to BAF and PBAF) is beyond the scope of the paper, but I will request that the authors revisit and discuss this issue in the Discussion to note its likely existence and possible participation in the data and processes.

*Reviewer #3:*

The polytenic chromosomes of *Drosophila* larval salivary glands allow the visualization of interphase chromosomes in a linear, extended form. The authors applied confocal life-imaging confocal microscopy and FRAP analysis to polytene chromosomes to study the dynamic interactions of the (P)BAP remodeling complexes with chromatin. They observe through a series of intriguing experiments that the BRM ATPase interacts with open chromatin only transiently in the presence of hydrolysable ATP. Rapid redistribution of the complex in the nucleus requires continuous ATP hydrolysis. Polytene squashes remove ATP and lead to artefactual trapping of the complex at the sites of otherwise transient interaction. The salivary glands can be cultivated for some time and remain active so that treatment with the ecdysone triggers the characteristic site-specific 'puffing' of the chromosome as a measure of transcription. The authors manage to subject individual 'puffs' to FRAP analyses, a nice demonstration of their technical proficiency.

Although the very transient interactions of other nucleosome remodeling factors have been reported by Erdel and Rippe, the case of BRM is of interest to the chromatin community. The imaging of polytene chromosomes in their squashed and 'life' state is particularly illuminating. The authors push the system and attempt to correlate histone turnover with the presence of the BAP complex.

The study presents a wealth of data in a very compact format. As a result, some arguments appear a bit abbreviated, some methods poorly explained and some controls are lacking for stringent argumentation. This interesting manuscript that can be significantly improved by addressing the following comments.

1. Many findings are illustrated by single individual pictures (e.g. most panels in figure 2). While these illustrate the observations well, they represent individual cells and lack statistical robustness. It will be reassuring if the authors mention at strategic places that all images are representative of the large majority of cells and that similar ones have been obtained in biological replicates (i.e. same experiment on different days). Should this not be the case the readers must be warned accordingly, and the type and source of heterogeneity be detailed.

2. The authors summarize published profiles to say that 70% of chromatin can be considered 'black'. Because they find only 1% of chromatin stained with GFP-H2B at 'black' intensity, they calculate a 25- to 150-fold level of chromatin compaction. Can we assume that the GFP-H2B intensities scale linearly with compaction? Since the GFP-histones are provided by transgenes, their expression is presumably not cell-cycle regulated. What are the levels of GFP-histones relative to the endogenous ones? Can we assume that the GFP-H2B is fully functional, i.e. a substrate to remodelers?

3. Figure 2 Supplement 1C. It is stated that GFP-BRM is efficiently incorporated into remodeler complexes. Here the level of GFP-BRM appears at least 10-fold higher that the level of endogenous BRM. This suggests that GFP-BRM is overexpressed to a worrying extent, where it may compete for the endogenous ATPase. How come that the excessive amount of BRM is not seen in the input? The question here is whether the properties of the tagged ATPase reflect those of the endogenous protein. Devil's advocate says that the endogenous protein constitutes the 'immobile fraction' that is, of course, not detected.

4. "Similar FRAP kinetics were observed for a low-expressing GFP-BRM *Drosophila* line (Figure 3—figure supplement 1A-C)." How is low expression defined? Given the overexpression of BRM (point 5 above), why was the analysis not done with a line with more physiological levels (since it was apparently available)?

5. It is said that "Depletion of BRM affects neither the polytene chromosome condensation pattern nor the binding of RNAPII (Figure 2—figure supplement 1A)". However, the shown image suggests otherwise as the level of pol II staining appears clearly diminished. If this statement was to be maintained some kind of quantification needs to be done.

6. The authors classify polytene banding as 'white', 'grey' and 'black' using defined fluorescence intensity thresholds. How does the live signal relate to the corresponding squashed chromosomes? How well do the histone-GFP values correlate to the DNA staining?

7. Figure 2 supplement 1D: Moira antibody appears to stain everything. How do we know the staining is specific?

8. Figure 2 supplement 1E lacks all controls (Western blots).

9. It is concluded that GFP-BRM is incorporated in (P)BAP and targeted correctly. What are the criteria for correct targeting and how has it been measured?

10. It is stated: "Pertinently, BRM and SNR1 levels within the polytene chromosome bands are much lower than in the nucleoplasm. This indicates that chromosome condensation leads to (P)BAP exclusion." This latter statement about causation cannot be derived from the correlative data.

11. What is the basis of the statement: "Note that loss of BRM does not affect polytene chromosome condensation (Figure 2-S1A)"? Is this just by looking at the image or has any other analysis been done? What is the sensitivity of the analysis – limitation of the statement?

12. It is stated "Pertinently, there was no appreciable difference in recovery kinetics between BRM in the nucleoplasm or associated with chromatin." What is the basis of this conclusion? Just looking at Figure 3A one would not be able to tell.

13. From Figure 3- supplement 1A-C one may conclude that the lower-expressing GFP-BRM binds better to the chromosome. Is this conclusion justified (and if not, why not?).

14. Figure 3C: which band was frapped for RPB3? Do the recovery curves also summarize the results of 17 band-FRAPs?

15. Related to the RPB3 FRAP profile it is stated "The rapid recovery at the start probably reflects dynamic binding to the promoter, whereas the linear phase corresponds to elongating RNAPII." This is pure speculation, should be removed from the 'results'.

16. The conclusion "We conclude that (P)BAP acts through a continuous and rapid "hit-and-run" probing of the genome" depends on a more stringent argumentation that the protein is functional (see above).

17. Figure 4f: Please indicate the number of biological replicates (assuming each biological replicate consists of three technical replicates).

18. Figure 4a shows the level of BRM recruitment to Ec-induced puffs. This is presumably fromn a wt larva. How does the same picture look like if the GFB-BRM overexpression line is used and stained for either GFP or BRM?

19. Figure 5C: the puffing is not visible, unlike stated in the text.

20. Figure 5H. "The absence of a full recovery of RNAPII at the height of E74 and E75 expression and puffing is similar to its behavior on fully induced heat shock loci 87A and 87C, and is indicative of local RNAPII recycling (Yao et al., 2007)." Might it also be possible that a certain fraction of pol II is stalled under culture conditions and thus 'trapped' in the topologically associated elongating form?

21. Figure 5J: Concluding about the absence of the D4 subunit requires some kind of a positive control and statistics.

22. Figure 6b: what is red-blue?

23. Figure 6c, 7c: How many bands were frapped?

24. "IF of polytene chromosome spreads revealed co-localization of GFP-BRM-K804R with endogenous MOR (SMARCC1/2), indicating largely normal targeting of this mutant". What are the criteria for 'Largely normal targeting' – what would be your criteria for 'not normal'?

25. Figure 7k. BRM binds interbands where histones are barely detectable. It is unclear how both proteins could be frapped at the same site. For H2B FRAP- was the neighbouring band frapped? What is the resolution of the assay?

26. The authors conclude that chromatin remodeling and remodeler relased are coupled. This conclusion is premature given the possibility of indirect effects (e.g. through elongating pol II -mediated by different transcription levels). Furthermore, the discussion states: "These observations suggest that the prevalent outcome of chromatin remodeling in vivo is restructuring or sliding of nucleosomes rather than eviction." The authors should acknowledge that they have very little spatial resolution: the histone dynamics could be very local. The data do not allow concluding about nucleosome remodeling mechanism.

27. The discussion states: "However, given that chromatin typically only occupies about 1.5% of the nuclear volume, …" Where does that number come from? That seems to be very odd.

---

## [Author Response]

Essential revisions:1) The study presents a wealth of data in a rather compact format. As a result, some arguments appear a bit abbreviated, some methods could be explained more thoroughly, and some controls are lacking for a more stringent argumentation. By addressing the corresponding reviewers' comments below, this highly interesting manuscript can be significantly improved.Although additional data could further strengthen the manuscript, we feel that the points raised by the reviewers can be adequately addressed with textual revision and by consolidating and more rigorously quantifying the data, and more explicitly stating controls, etc. Note that no new experimentation is called for.

We thank the reviewers for their constructive comments and suggestions, which have helped us to improve our manuscript. We now provide a more extensive and explicit explanation of the quantification of the polytene nuclei images in the legends and methods, which were actually extremely consistent across different animals and different nuclei in the salivary glands. For all other experiments quantitative information was already provided. In addition, we have reorganized the manuscript according to the reviewers suggestions, and changed the text to increase clarity. Finally, we have included additional data (detailed below) to clarify some of the concerns of reviewer #3.

2) One worry is that the tagged proteins are not functional, but there may be good arguments as to why this is less of an issue (e.g., the effect of ATP depletion and KR mutation). The authors should make these arguments in the paper.

Tagging may indeed affect a protein’s function. However, all major conclusions derived from tagged factors (such as remodeler distribution, the effects of ATP depletion, ATP add-back and the BRM-K804R mutation) were confirmed by IF of endogenous (P)BAP subunits. These effects were entirely consistent across multiple (P)BAP subunits. Moreover, other tagged chromosomal regulators (RNAPII, Polycomb, HP1, EcR) that were analyzed in parallel to (P)BAP behaved differently. We have expanded these arguments in the text.

3) All three reviewers felt that the clarity of the presentation could be improved by bringing together the sections related to ATP binding and hydrolysis. We strongly recommend such a change in the order of presentation.

We thank the reviewers for this valuable suggestion. We have completely reorganized the manuscript. We now first present our results on chromosomal organization and (P)BAP intranuclear distribution. Next, we present the data on the role of ATP in remodeler dynamics. The following changes were made (new<old): Figure 1<Figure 2, Figure 2<Figure 4, Figure 3<Figure 5A-D, Figure 4<Figure 1, Figure 5<Figure 3, Figure 6<Figure 5F-K, Figure 7<Figure 6, Figure 8<Figure 7, Figure 9<Figure 8. To clarify some issues raised by reviewer #3, we have provided new data in Figure 2C, Figure 3C, Figure 6A-B, Figure 5-suppl.1G-H, and Figure 6-suppl.1C-D.

Reviewer #1:Strength: Chromatin remodelers play critical roles in mobilization of nucleosomes to modulate chromatin-templated functions, but the molecular mechanisms underlying the dynamics of remodelers interacting with chromatin remains incompletely understood. By using cellular imaging, this study reports that chromatin remodeler mutant, which is defective in ATP-binding, reside longer at chromatin than wild-type one, and further indicates that ATP-hydrolysis stimulates remodeler release from chromatin. By leveraging the characteristic of polytene chromatin being visible under fluorescence microscopy, the authors study that the dynamics of chromatin remodelers at native loci, providing further evidence that chromatin remodelers dynamically interact with chromatin with turn-over time less than 5 sec. Overall, these studies indicate that chromatin remodelers rapidly probe the genome and ATP-binding and hydrolysis play important roles in release of remodelers from chromatin.Weakness: There are a few studies that have reported that remodelers dynamically exchange with chromatin. The novelty of this study is the discovery of the roles of ATP-binding and hydrolysis on the dissociation of remodelers from chromatin; however, to this reviewer, there is lack of functional significance of the destabilization or mechanistic insights into the destabilization.

Our work shows that remodeling and release are mechanistically coupled. This is significant because it shows that the remodeling process is intrinsically dynamic and does not induce a stable static state. Moreover, we show that during remodeling, remodelers turn over faster than histones. An open chromatin state requires continuous remodeler activity and allows for quick regulation. The use of polytene nuclei enabled us to visualize these processes on endogenous, natural gene loci in living cells. We show that induction of *E74/75* depends on PBAP. Thus, we are studying PBAP and chromatin organization in living tissue on physiological target genes that depend on PBAP for its regulation. This has not been done previously.

1) The authors study the dynamics of BRM-K804R within polytene nuclei (Figure 7). The authors also investigate the dynamics of wild-type BRM at the E74 and E75 loci (Figure 5). It appears to this reviewer that one nice experiment would be to investigate the exchange dynamics of BRM-K804R at these two loci and to study the effects of BRM-K804R on their transcriptional levels. Thus, it would be possible to correlate the binding dynamics and transcription and to provide potential functional significances of rapid exchange of remodelers at chromatin.

BRM-K804R clamps to chromosomes and is also completely immobile on E74/75. As first shown by Tamkun and colleagues (Armstrong et al., 2002), BRM-K804R is a dosagedependent dominant mutant that interferes with all RNAPII transcription, including from the E74/75 locus. We have now included results showing that BRM kinetics on RNAPII loci is indistinguishable from that on non-transcribed loci (Figure 5-suppl.1G-H)

2) To this review, it is not easily to follow the logic of the manuscript. It would be helpful to readers if the authors give sufficient reasonings for some experiments. For instance, why do the authors study PBAP and BAP for Ec-induced gene expression (Page 12). To me, the authors should also reorganize some figures and change the order of the presentation.

We have reorganized the manuscript according to the recommendations of the reviewers. The importance of the experiments on (P)BAP and Ec-induced transcription is that these provide the foundation to study PBAP dynamics by life imaging on genes that dependent on that remodeler in vivo.

3) I would suggest that the authors quantify FRAP data to extract kinetic parameters based kinetic modelling. In the current version, when comparing t_1/2_ between different cellular systems, it is not informative.

We have used the half time of recovery and the (im-)mobile fraction to compare the kinetics of the interactions of individual (P)BAP subunits as well as with other known chromatin binding proteins. In addition, we used these parameters to describe the differences in chromatin interaction between WT and mutant BRM. To extract additional kinetic parameters by modelling, a (currently unavailable) model containing detailed information about the interaction(s) between (P)BAP and chromatin is required. Moreover, we refrain from kinetic modelling on our FRAP data because these types of analyses can easily lead to over- or misinterpretations (discussed by e.g., Mueller et al., Curr. Opin. Cell Biol. 23, 403-411, 2010). Importantly, our FRAP data provides an upper limit to (P)BAP residence time and establishes that (P)BAP has a very fast turnover, whereas histones are comparatively stable. Additional kinetic data on remodeler/chromatin interactions are reported in the accompanying study by Kim et al. using single molecule analysis, a system more suitable for detailed kinetic analysis. The findings of Kim et al. (https://doi.org/10.1101/2021.04.21.440742) agree well with our results. We have included a reference to this manuscript.

4) 1,6-hexanediol has been widely used in phase-separation studies. It primarily disrupts hydrophobic interactions. Given that LLPS involves multiple non-covalent interactions, the authors should re-phase some of their conclusions.

We have included the 1,6-HD experiments because every time we presented these data in a seminar, someone asked about LLPS and 1,6-HD. So, we used it simply as an additional test that provides further evidence that LLPS appears to play no substantial role in the effects we describe in this paper.

5) A schematic that describes PBAP and BAP would help non-remodeler readers in Figure 1.

We have added cartoons of BAP and PBAP in Figure 2C.

(6) Figure 1e shows difference in FRAP curves between wild-type BRM and BRM-K804R. BRM-K804R forms clusters while BRM not. It would be helpful if FRAP of BRM-K804R is also done in cluster and no-cluster regions.

This would indeed be a very interesting experiment. In this figure, the kinetics of GFP-BRM and GFP-BRM-K804R were studied by strip FRAP on *Drosophila* S2 cells. Because of the size of the individual nuclei (Ø < 4 µm), a 16-pixel wide strip was bleached in these experiments. Due to limitations in resolution, discriminating between clustered and non-clustered BRM-K804R was not possible.

7) Page 6, the last conclusion sentence seems over-stated since BRM-K804R impacts ATP-binding rather than hydrolysis.

We have re-phrased this sentence (p10, line 4).

8) Page 10, the second to last sentence, there is no data to support the author's claim.

The bleached area in the FLIP-FRAP experiments presented in figure 5A consists of both nucleoplasmic and chromatin-associated GFP-BRM. As can be concluded from these images, recovery of both compartments occurs at a similar rate. We have re-phrased this sentence (p10, line 23).

9) Page 12, the conclusion remark of the first paragraph, there is no data to support hit-and-run' model.

We removed “hit-and-run”.

10) To this reviewer, it appears that the retention of BRM-K804R at chromatin at polytene chromatin cannot be simply explained by a slow dissociation rate and a fast on-rate. This is because BRM-K804R can trap wild-type BRM at polytene chromatin and BRM-K804R can be recovery rapidly in S2 cells. Alternative interpretation should be given.

We have modified our discussion on the difference between polytene nuclei and S2 cells for greater clarity (p16, line 14). Although this remains speculative, we propose that the high local density of (P)BAP loci in polytene chromosomes amplifies the difference in retention time between wtBRM and BRM-K804R.

Reviewer #2:ATP-dependent chromatin remodelers must scan the genome for locations to open chromatin (to help provide access to transcription and DNA repair factors) via nucleosome movement or eviction, and then release from that location to conduct new work. Currently, it is challenging to quantify remodeler-nucleosome dynamics in vivo, and the role of ATP binding and hydrolysis on the retention and recycling of remodelers at nucleosomes – aside from the known role of ATP in the process of DNA translocation. These questions are of general interest as chromatin remodelers are intimately involved in the regulation of DNA accessibility for processes such as transcription and DNA repair, and as remodelers must both associate with regions productively yet release properly to search for and conduct the scale of work needed.Here, Tilly et al., (Verrijzer Lab) and colleagues address these questions using elegant live-cell imaging, applied to monitor the SWI/SNF-family Brahma (BRM) remodeler in *Drosophila* polytene nuclei. This method combined with cleverly chosen reagents, mutants, or physiological contexts, allowed the authors to measure chromatin condensation along with the dynamics governing the interaction of BRM with chromatin. Several conclusions of interest could be reached through the work:– With the vast majority of BRM is unbound at any given time, BRM remodeler interacts very transiently with chromatin, browsing nucleosomes largely located in decondensed regions.– ATP binding reduces BRM retention at chromatin, as revealed by characterizing the in vivo dynamics of a Walker A motif mutant.– ATP hydrolysis increases BRM release from chromatin, as revealed by characterizing the in vivo dynamics in the presence of ADP, or a slow hydrolysable ATP analogue, or a non-hydrolysable ATP ground state mimic.– When applied to the specific physiological context of highly transcribed ecdysone-induced loci, only the (BRM-containing) PBAP remodeler appears essential; and, remarkably, PBAP fulfills this function without a local retention as shown for RNAPII, but by a rapid exchange.– Consistent with – and expanding on – the previous conclusions, the presence of the ATP-binding deficient BRM mutant leads to the clustered binding of not only mutated BRM but also wild-type BRM, without the involvement of a phase separation (as 1,6-hexanediol did not affect the punctate/clustered pattern) but likely by accumulation due to excessive retention. Notably, as the authors rightly comment, these interesting results lead to the possibility of a direct cooperation or a hand-off mechanism between different tandem SWI/SNF remodelers, and will deserve further future investigation.The conclusions listed above are relatively convincingly as they are well supported by the data. The discussion is clearly and thoughtfully written.Altogether, this work rigorously addresses how ATP binding and hydrolysis cycle of the Brm remodeler, affects the in vivo dynamics of nucleosome browsing, binding and recycling. I think this will be of considerable interest to the remodeler and chromatin community, and recommend publication after revision in eLife.1. To improve the manuscript, I will suggest that the authors consider slightly reorganizing the manuscript in order to bring in close proximity all the sections related to ATP binding and to ATP hydrolysis, which are currently separated by the characterization of the BRM dynamics at the ecdysone-induced loci.

As detailed above, we have reorganized the manuscript according to the recommendations of the reviewers.

2. I will also ask that the authors note that the reorientation of the two ATPase lobes is not just induced by nucleosomes, but also efficiently elicited by naked DNA, so nucleosomes do not gate that conformational change.

We have corrected the text (p17, line 20).

3. I appreciate how the kinetics of the in vivo work was linked to former in vitro studies in the Discussion – and here the authors might consider (in addition to the works cited) how prior pre-steady state stopped-flow kinetic analysis of SWI/SNF-family remodelers and resulting kinetic and probabilistic models (Fischer et al., Biochemistry 2007) also generally align with their work – and how thinking about release as a probability during each ATPase cycle might be of interest.

We have included these prescient findings in our discussion and included the insightful notion of release as a probability during each ATPase cycle (p18, line 7).

4. Finally – the identification of a GLTSCR1 ortholog (Figure 1) is really interesting. he authors mention this but do not explore more deeply. This affects the work in two ways – the possible existence of a GBAF/ncBAF related complex in flies, and that this complex may be participating in the processes observed. I understand that a full characterization of a possible separate fly GBAF/ncBAF complex (in addition to BAF and PBAF) is beyond the scope of the paper, but I will request that the authors revisit and discuss this issue in the Discussion to note its likely existence and possible participation in the data and processes.

We have corrected the discussion and included a reference to a recent publication from the Bellen lab (Barish et al., 2020) that includes a further characterization of GBAF/ncBAF in *Drosophila* (page 14, line 9).

Reviewer #3:The polytenic chromosomes of *Drosophila* larval salivary glands allow the visualization of interphase chromosomes in a linear, extended form. The authors applied confocal life-imaging confocal microscopy and FRAP analysis to polytene chromosomes to study the dynamic interactions of the (P)BAP remodeling complexes with chromatin. They observe through a series of intriguing experiments that the BRM ATPase interacts with open chromatin only transiently in the presence of hydrolysable ATP. Rapid redistribution of the complex in the nucleus requires continuous ATP hydrolysis. Polytene squashes remove ATP and lead to artefactual trapping of the complex at the sites of otherwise transient interaction. The salivary glands can be cultivated for some time and remain active so that treatment with the ecdysone triggers the characteristic site-specific 'puffing' of the chromosome as a measure of transcription. The authors manage to subject individual 'puffs' to FRAP analyses, a nice demonstration of their technical proficiency.Although the very transient interactions of other nucleosome remodeling factors have been reported by Erdel and Rippe, the case of BRM is of interest to the chromatin community. The imaging of polytene chromosomes in their squashed and 'life' state is particularly illuminating. The authors push the system and attempt to correlate histone turnover with the presence of the BAP complex.The study presents a wealth of data in a very compact format. As a result, some arguments appear a bit abbreviated, some methods poorly explained and some controls are lacking for stringent argumentation. This interesting manuscript that can be significantly improved by addressing the following comments.1. Many findings are illustrated by single individual pictures (e.g. most panels in figure 2). While these illustrate the observations well, they represent individual cells and lack statistical robustness. It will be reassuring if the authors mention at strategic places that all images are representative of the large majority of cells and that similar ones have been obtained in biological replicates (i.e. same experiment on different days). Should this not be the case the readers must be warned accordingly, and the type and source of heterogeneity be detailed.

IF images of polytene spreads and whole mount images of salivary glands (or imaginal disks or embryos) are normally highly reproducible, therefore we did not state this explicitly. However, we have now included statements on the reproducibility of images of individual nuclei in the legends to Figure 1, 7 and 8. More importantly, we note that the IF images of different PBAF subunits in different figures are very similar. Moreover, IF and live cell imaging of polytene nuclei yield very similar images. Collectively, these comparisons emphasize the reproducibility of the images presented. In response to some later criticisms from this reviewer, we have included additional data (Figure 3C, Figure 6A-B, Figure 5-suppl.1G-H, and Figure 6-suppl.1C-D), which further illustrates the reproducibility of our imaging.

2. The authors summarize published profiles to say that 70% of chromatin can be considered 'black'. Because they find only 1% of chromatin stained with GFP-H2B at 'black' intensity, they calculate a 25- to 150-fold level of chromatin compaction. Can we assume that the GFP-H2B intensities scale linearly with compaction? Since the GFP-histones are provided by transgenes, their expression is presumably not cell-cycle regulated. What are the levels of GFP-histones relative to the endogenous ones? Can we assume that the GFP-H2B is fully functional, i.e. a substrate to remodelers?

1) All data were collected using conditions that preserved linearity during data acquisition as much as possible. However, we cannot exclude small deviations at very low and very high fluorescence intensities. Moreover, there might be shielding and other effects that interfere with strict linearity. The GFP-H2B fluorescence was quantitated from 10 individual nuclei, 9 slices/nucleus. From this data set we concluded that 5% (not 1%) of the chromatin volume is “black” and 43% is “grey”. As explained in the methods, the chromocenter and telomers are recognizable structures that were used to calibrate the signal from black chromatin. Likewise, interbands and puffs were used to calibrate white chromatin. Chromatin bands that fall between “white” and “black” were assigned grey. This classification does not depend on linear scalability of GFP-H2B. The goal of these experiments was not to provide a very precise estimate of the amount of white, grey, and black chromatin, but to obtain an impression of their distribution in living cell polytene nuclei. What we find amazing is how well our estimates derived from vital imaging agree with those of Eagen et al. (2015), who used a completely different method (Hi-C) relying on fixed cells.

2) GFP-H2A and mCh-H2B fully co-localize (Figure 1F) and recapitulate the canonical chromosomal banding pattern. We cannot detect appreciable amounts of free histones in the nucleoplasm or cytoplasm, indicating there is no substantive over-expression. Moreover, the *Drosophila* expressing the tagged histones develop, pupate, and propagate normally. Thus, we have no indications that these tagged histones are not behaving normally.

3. Figure 2 Supplement 1C. It is stated that GFP-BRM is efficiently incorporated into remodeler complexes. Here the level of GFP-BRM appears at least 10-fold higher that the level of endogenous BRM. This suggests that GFP-BRM is overexpressed to a worrying extent, where it may compete for the endogenous ATPase. How come that the excessive amount of BRM is not seen in the input? The question here is whether the properties of the tagged ATPase reflect those of the endogenous protein. Devil's advocate says that the endogenous protein constitutes the 'immobile fraction' that is, of course, not detected.

We cannot detect BRM or GFP-BRM in the input extract from dissected salivary glands by western blotting. This is probably due to interfering proteins in that size range. BRM and GFPBRM are only detected in the MOR co-IPs, thus by definition as part of a remodeler complex. Note that we have no problems detecting GFP-SNR1, SNR1 and MOR. Loss of the architectural subunit MOR, leads to disintegration of the (P)BAP complexes and loss of GFPBRM (see Figure 1.suppl.1E, and Moshkin et al., 2007). This observation argues strongly that GFP-BRM only exists as part of a complex and not as a free protein. This interpretation is also in line with the published literature (see e.g., Mashtalir et al., 2018 Cell 175, p1272). Most likely, free endogenous BRM is degraded because its place in the (P)BAP complex is taken by GFP-BRM (see also Figure 4A). Importantly, all our major conclusions are supported by analysis of endogenous proteins and do not rely solely on the tagged remodeler subunits. Finally, the FRAP kinetics of GFP-SNR1, GFP-D4 and GFP-BRD7 are all very similar to those of GRPBRM (including the low expressing line).

4. "Similar FRAP kinetics were observed for a low-expressing GFP-BRM *Drosophila* line (Figure 3—figure supplement 1A-C)." How is low expression defined? Given the overexpression of BRM (point 5 above), why was the analysis not done with a line with more physiological levels (since it was apparently available)?

The primary reason is that we generated the Sgs3>GFP-BRM line before the #59784 fly line became available. Expression of GFP-BRM in the salivary gland nuclei of the 59784 line is very low compared to Sgs3>GFP-BRM, as judged by fluorescence. Importantly, nuclear distribution and kinetics of GFP-BRM in #59784 nuclei is indistinguishable from that in GFPBRM in Sgs3>GFP-BRM. Thus, our measurements are not influenced by the expression level of GFP-BRM.

5. It is said that "Depletion of BRM affects neither the polytene chromosome condensation pattern nor the binding of RNAPII (Figure 2—figure supplement 1A)". However, the shown image suggests otherwise as the level of pol II staining appears clearly diminished. If this statement was to be maintained some kind of quantification needs to be done.

We respectfully disagree. The RNAPII and DAPI stains of polytenes from the wt and BRM knockdown line both fall within the normal range of variability we observe for polytene spreads (whereas BRM is no longer detectable). Thus, we do not detect a gross change in the polytene banding pattern or RNAPII binding. These assays are qualitative rather than quantitative. We have changed the text for greater clarity (p5, line 19).

6. The authors classify polytene banding as 'white', 'grey' and 'black' using defined fluorescence intensity thresholds. How does the live signal relate to the corresponding squashed chromosomes? How well do the histone-GFP values correlate to the DNA staining?

The correspondence is so good that we can use the polytene maps to navigate the chromosomes in a living nucleus and identify the chromocenter, telomers, and specific loci like *E74/75* in living gland nuclei.

7. Figure 2 supplement 1D: Moira antibody appears to stain everything. How do we know the staining is specific?

MOR/(P)BAP binds many interband loci. The anti-MOR antibody is specific and validated previously by depletion and IP-mass spectrometry. See e.g., Moshkin et al., 2007; Chalkley et al., 2008 and Mohrmann et al., 2004.

8. Figure 2 supplement 1E lacks all controls (Western blots).

The absence of GFP-BRM in the salivary gland polytene nuclei from MOR^KD^ larvae is clear. As mentioned above, after blotting of extracts from dissected salivary glands (which is not trivial), BRM is not detectable, possibly due to other proteins in this range. The observation that MOR is a key architectural subunit (with all control blots) has been published previously (Moshkin et al., 2007).

9. It is concluded that GFP-BRM is incorporated in (P)BAP and targeted correctly. What are the criteria for correct targeting and how has it been measured?

As described in the results, we concluded that GFP-BRM is incorporated into (P)BAP and targeted correctly based on:

1) Mass spectrometric analyses of anti-GFP IPs (Figure 4B).

2) Western blots of MOR-IPs, using anti-GFP, anti-BRM, anti-SNR1 and anti-MOR antibodies (Figure 1suppl.1C).

3) GFP-BRM expression in MOR^KD^ larvae (Figure 1suppl.1E).

4) Comparison of GFP-BRM distribution on polytene chromosomes with MOR

(Figure 1suppl.1D).

5) Comparison of the nuclear distribution of GFP-BRM with endogenous PBAP (compare Figure 1B and I).

6) GFP-BRM binds canonical BRM target loci (Figure 3B, Figure 4F)

10. It is stated: "Pertinently, BRM and SNR1 levels within the polytene chromosome bands are much lower than in the nucleoplasm. This indicates that chromosome condensation leads to (P)BAP exclusion." This latter statement about causation cannot be derived from the correlative data.

Although we do not formally prove causation, (P)BAP is excluded from condensed polytene chromatin bands, whereas (P)BAP is not required to form interbands (Fig1suppl.1A). Thus, a parsimonious explanation would be that a high level of chromatin condensation causes the exclusion of (P)BAP. Nevertheless, we have re-phrased this statement to reflect this notion is a possibility, not a conclusion (p7, line 4; p14, line 23).

11. What is the basis of the statement: "Note that loss of BRM does not affect polytene chromosome condensation (Figure 2-S1A)"? Is this just by looking at the image or has any other analysis been done? What is the sensitivity of the analysis – limitation of the statement?

Yes, this is based on inspection of images, which showed that in the absence (P)BAP the classical banding pattern of polytene chromosomes is not changed appreciably. Analysis of polytene chromosomes allows for an easy “whole genome” analysis of global chromosome structure. It is not a high-resolution technique. However, the work of Eagen and colleagues (2015), established a close correspondence between the banding pattern of polytene chromosomes and TADs.

12. It is stated "Pertinently, there was no appreciable difference in recovery kinetics between BRM in the nucleoplasm or associated with chromatin." What is the basis of this conclusion? Just looking at Figure 3A one would not be able to tell.

The bleached area in the FLIP-FRAP experiments presented in new Figure 5A consists of both nucleoplasmic and chromatin-associated GFP-BRM. As can be concluded from these images, recovery of both compartments occurs at a similar rate. We have re-phrased this sentence (p10, line 22).

13. From Figure 3- supplement 1A-C one may conclude that the lower-expressing GFP-BRM binds better to the chromosome. Is this conclusion justified (and if not, why not?).

In our opinion, sg>GFP-BRM and line GFP-BRM_low_ show comparable distributions.

Importantly, quantitative FRAP experiments confirmed this observation. The results from the FRAP experiments are very comparable with respect to t_½_ ( GFP-BRM_low_ , t_½_ = 4.2 ± 1.6 s, n=15 and GFP-BRM, t_½_ = 3.1 ± 1.1, n=17) and mobile fraction: 0.9 ± 0.1 for both GFP-BRM_low_ and GFP-BRM.

14. Figure 3C: which band was frapped for RPB3? Do the recovery curves also summarize the results of 17 band-FRAPs?

New Figure 5G shows a representative single FRAP curve for RBP3 (we have performed multiple experiments in different lines). The kinetics we observe for both RBP3 and RBP2 (Figure 6E) is very similar to those obtained earlier by the Lis lab, using different lines, and analyzing different loci. We refer to the extensive analysis of RNAPII by Lis et al., in our results. As we basically confirm their earlier results, we see no need to expand the RNAPII analysis, which functions as a reference for (P)BAP.

15. Related to the RPB3 FRAP profile it is stated "The rapid recovery at the start probably reflects dynamic binding to the promoter, whereas the linear phase corresponds to elongating RNAPII." This is pure speculation, should be removed from the 'results'.

Again, we refer to the rigorously tested conclusions published earlier by Lis and colleagues, and indicate that our RNAPII FRAP curves are highly similar.

16. The conclusion "We conclude that (P)BAP acts through a continuous and rapid "hit-and-run" probing of the genome" depends on a more stringent argumentation that the protein is functional (see above).

As discussed above, our conclusions are fully supported by results obtained with the endogenous (P)BAP. We removed “hit-and-run”.

17. Figure 4f: Please indicate the number of biological replicates (assuming each biological replicate consists of three technical replicates).

We used 3 independent biological replicates. This information was added to the legends.

18. Figure 4a shows the level of BRM recruitment to Ec-induced puffs. This is presumably fromn a wt larva. How does the same picture look like if the GFB-BRM overexpression line is used and stained for either GFP or BRM?

Unless stated otherwise, *Drosophila* images and data shown are derived from wild type animals. We use the GFP-BRM for life cell imaging and it binds the same loci as the endogenous protein (see Figures 2, 3 and 5).

19. Figure 5C: the puffing is not visible, unlike stated in the text.

We have included a lower magnification of the same nucleus to show the zoom area. In our opinion this is a clear puff with a diameter much larger than that of a regular polytene region (Figure 3C). This is confirmed by direct measurements of chromosomal areas (Figure 3E).

20. Figure 5H. "The absence of a full recovery of RNAPII at the height of E74 and E75 expression and puffing is similar to its behavior on fully induced heat shock loci 87A and 87C, and is indicative of local RNAPII recycling (Yao et al., 2007)." Might it also be possible that a certain fraction of pol II is stalled under culture conditions and thus 'trapped' in the topologically associated elongating form?

We cannot exclude this possibility. Therefore, we referred to the results by Yao et al. (2007), who made similar observations on the HS loci and provided additional evidence that RNAPII was recycled, not stalled. In our study, the kinetics of GFP-RPB2 was studied 30 min after ecdysone addition, at the peak of RNAPII accumulation and induction of E74/75, making it unlikely this is stalled RNAPII. However, we have modified the text to indicate this possibility (p11, line 21).

21. Figure 5J: Concluding about the absence of the D4 subunit requires some kind of a positive control and statistics.

We have included a GFP-D4 image showing a full nucleus (Figure 6.suppl.1D).

22. Figure 6b: what is red-blue?

The false color scale bar represents full scale grey values (8 bit) from 0 (black) to 255 (white). This information was added to the legend.

23. Figure 6c, 7c: How many bands were frapped?

Figure 6C, at least 6; Figure 7C, at least 8. These FRAP curves are highly invariable across different animals.

24. "IF of polytene chromosome spreads revealed co-localization of GFP-BRM-K804R with endogenous MOR (SMARCC1/2), indicating largely normal targeting of this mutant". What are the criteria for 'Largely normal targeting' – what would be your criteria for 'not normal'?

This figure shows the strong co-localization of GFP-BRM-K804R and endogenous MOR, indicated by the predominantly yellow merge, and by zooming, which shows a very similar pattern for GFP and MOR. Clearly separated red and green in the merge would indicate not normal targeting of GFP-BRM-K804R.

25. Figure 7k. BRM binds interbands where histones are barely detectable. It is unclear how both proteins could be frapped at the same site. For H2B FRAP- was the neighbouring band frapped? What is the resolution of the assay?

Bleached area is 4 x 2 µm (there is a scale bar in this panel). Thus, we do not reach band/interband resolution. Nevertheless, we observe a clear and significant reduction in histone turnover. We also note that although BRM or GFP-BRM levels are low in bands they are not completely absent (see e.g., Figure 1B,C and Figure 1.suppl.1F).

26. The authors conclude that chromatin remodeling and remodeler relased are coupled. This conclusion is premature given the possibility of indirect effects (e.g. through elongating pol II -mediated by different transcription levels). Furthermore, the discussion states: "These observations suggest that the prevalent outcome of chromatin remodeling in vivo is restructuring or sliding of nucleosomes rather than eviction." The authors should acknowledge that they have very little spatial resolution: the histone dynamics could be very local. The data do not allow concluding about nucleosome remodeling mechanism.

1) Remodeling and release are coupled because both depend on ATP-hydrolysis. The possibility that there might be indirect effects does not change that.

2) There are many (P)BAP-binding loci (actually the majority) that do not coincide with sites of active transcription. We did not observe a difference in BRM dynamics between transcribed versus non-transcribed loci. We have now added these results (Figure 5.suppl.1G,H).

3) In our discussion, we first review and cite results from other labs, before we suggest that histone eviction is not the prevalent outcome of in vivo remodeling. Clearly we do not solely rely on our own results.

4) We explicitly state that “ … histone eviction is restricted to small regulatory regions of the genome and is associated with the placement of variant histones (Brahma and Henikoff, 2020; Cakiroglu et al., 2019; Deal et al., 2010; Pillidge and Bray, 2019).”

5) Nowhere in the manuscript do we state that there is no histone turnover. In fact, we quantify histone turnover in Figure 5H and Figure 8K.

27. The discussion states: "However, given that chromatin typically only occupies about 1.5% of the nuclear volume, …" Where does that number come from? That seems to be very odd.

The diameter of a typical human nucleus is 10 μm (r=5 μm). V=4/3πr^3, yielding a volume of ~420 μm^3.

Human genome: diploid genome ~ 6.10^9 bp. A fully packaged genome, assuming 39 bp linker DNA between nucleosomes would be roughly 80% nucleosomal and 20% linker DNA. Volume bp ~ 1nm^3. Thus volume linker DNA: 0.2 x ~ 6.10^9 x bp ~ 1nm^3 = ~1.2 μm^3. The shell volume of a nucleosome is 3.28e+5 Å^3. nucleosomal genome: 0.8 x 6.10^9 / 147 = 3.3.10^7 nucleosomes, resulting in an approximate volume of: 0.33e+8 x 3.28e+2 nm^3 = 1.1e+10 nm^3 = 11 μm^3

Total volume chromatin: 1.2+11 = 12.2 μm^3. This corresponds to ~2.9% of the nuclear volume.

If the human genome was not packaged into chromatin.

Human genome: diploid genome ~ 6.10^9 bp, Volume bp ~ 1nm^3, thus human genome occupies ~6 μm^3. This corresponds to ~1.4% of the nuclear volume.